# Protein quality control and regulated proteolysis in the genome-reduced organism *Mycoplasma pneumoniae*

Raul Burgos[1] (iD), Marc Weber[1] (iD), Sira Martinez[1] (iD), Maria Lluch-Senar[1] (iD) & Luis Serrano[1,2,3,*] (iD)

## Abstract

**Protein degradation is a crucial cellular process in all-living systems. Here, using *Mycoplasma pneumoniae* as a model organism, we defined the minimal protein degradation machinery required to maintain proteome homeostasis. Then, we conditionally depleted the two essential ATP-dependent proteases. Whereas depletion of Lon results in increased protein aggregation and decreased heat tolerance, FtsH depletion induces cell membrane damage, suggesting a role in quality control of membrane proteins. An integrative comparative study combining shotgun proteomics and RNA-seq revealed 62 and 34 candidate substrates, respectively. Cellular localization of substrates and epistasis studies supports separate functions for Lon and FtsH. Protein half-life measurements also suggest a role for Lon-modulated protein decay. Lon plays a key role in protein quality control, degrading misfolded proteins and those not assembled into functional complexes. We propose that regulating complex assembly and degradation of isolated proteins is a mechanism that coordinates important cellular processes like cell division. Finally, by considering the entire set of proteases and chaperones, we provide a fully integrated view of how a minimal cell regulates protein folding and degradation.**

**Keywords** ATP-dependent protease; mycoplasma; protein degradation; proteomic approach; regulated proteolysis

**Subject Categories** Microbiology, Virology & Host Pathogen Interaction; Post-translational Modifications & Proteolysis

**Mol Syst Biol. (2020) 16: e9530**

## Introduction

Protein degradation is a key biological process that shapes the proteome of cells in response to internal and external signals. During stress conditions, misfolded and damaged proteins may accumulate in the cell with potential harmful consequences. In this scenario, chaperones and proteases play a key role as protein quality control factors assisting and reverting this situation through protein refolding or degradation (Mogk *et al*, 2011). Apart from maintaining protein homeostasis, protein degradation is also an efficient mechanism to induce changes in cell physiology (Van Melderen & Aertsen, 2009). Indeed, regulated proteolysis also occurs on intact proteins, like transcription factors or regulatory proteins responsible for physiological transitions (Mahmoud & Chien, 2018). Therefore, protein degradation can potentially regulate many biological processes depending on the variety of proteases and the possible substrates existing in a cell system.

In bacteria, most intracellular proteolysis is mediated by the ATP-dependent proteases belonging to the AAA+ family proteins (Bittner *et al*, 2016), including ClpXP, Lon, and FtsH (common to both gram-positive and gram-negative bacteria), ClpAP and ClpYQ (only in gram-negative bacteria), and ClpCP and ClpEP (only in gram-positive bacteria). All of these form oligomeric structures with two functional domains: an ATPase domain with unfolding and translocating activities that is dependent on ATP-hydrolysis, and a protease domain with proteolytic activity. The Clp family of proteases is characterized to encode these domains in separate polypeptides, whereas Lon and FtsH have these domains encoded in the same protein sequence. Notably, FtsH is a membrane-anchored protein and the only essential protease in *Escherichia coli* (Ogura *et al*, 1999).

Protein half-lives range from minutes to days, reflecting distinct degradation rates that generally are determined by the presence of degradation signals that trigger protease engagement. A degradation signal is normally referred to as a degron and can be intrinsically present in a protein or added by post-translational modifications. A common mechanism in both eukaryotes and prokaryotes is the N-end rule pathway, in which the identity of the N-terminal amino acids determines the half-life of the protein (Mogk *et al*, 2007). In prokaryotes, N-degrons are recognized by the protein adaptor ClpS, which delivers the substrate to the ClpAP protease (Erbse *et al*, 2006; Schmidt *et al*, 2009). N-degrons can also be created post-translationally, for example, by exposing destabilizing residues after proteolytic cleavage, or by adding Leu and Phe destabilizing

1  Centre for Genomic Regulation (CRG), The Barcelona Institute of Science and Technology, Barcelona, Spain
2  Universitat Pompeu Fabra (UPF), Barcelona, Spain
3  ICREA, Barcelona, Spain
   *Corresponding author. Tel: +34 93 3160101; E-mail: luis.serrano@crg.eu

residues to the N-terminus through the leucyl/phenylalanyl-tRNA-protein transferase (Ninnis *et al*, 2009). Similarly, adding a C-terminal tag by the bacterial tmRNA system promotes degradation of the modified protein (Keiler *et al*, 1996). Regardless of the mechanism, proteolysis is an energy-expensive and irreversible process and therefore must be strictly regulated. This is especially true for intracellular proteases, for which substrate recognition is very selective. In this respect, protein adaptors modulate substrate specificity, such as SspB, which recognizes and delivers tmRNA-tagged substrates to the ClpXP protease in *E. coli* (Levchenko *et al*, 2000; Kuhlmann & Chien, 2017). The accessibility of a degradation motif is also an important determinant factor regulating proteolysis. For instance, hydrophobic residues tend to be buried in the interior of a native protein but become exposed in misfolded proteins. This fact has been proposed to provide a discrimination factor to identify poor-quality proteins, as Lon and other quality control proteins tend to interact with hydrophobic regions (Rudiger, 1997; Chen & Sigler, 1999; Patzelt *et al*, 2001; Gur & Sauer, 2008). Even though substrate recognition is a common step for proteases, previous studies using substrate trapping methods have revealed that the substrate repertoire of cognate proteases varies within distinct prokaryotic systems and that the nature of the degradation signal seems to be diverse and generally sequence-independent (Liao & van Wijk, 2019). Conversely, there are also examples of substrate overlapping among distinct proteolytic systems, suggesting in some cases common mechanisms of recognition (Kuo *et al*, 2004; Tsilibaris *et al*, 2006; Lies & Maurizi, 2008).

Most of the studies mentioned above have been performed in complex model organisms such as *E. coli*, *Bacillus subtilis*, *Staphylococcus aureus*, and *Caulobacter crescentus* (Flynn *et al*, 2003; Feng *et al*, 2013; Bhat *et al*, 2013; Trentini *et al*, 2016; Arends *et al*, 2016, 2018). As the complexity of these organisms makes it difficult to have an integrated view of the protein degradation machinery, we chose to study this machinery in the genome-reduced bacterium *Mycoplasma pneumoniae,* an important human pathogen that causes community-acquired pneumonia, and is considered to be one of the smallest known self-replicating organisms. In the past years, a large effort has been made to obtain and integrate large "omics" datasets to quantitatively understand the biology of this minimal cell model (Güell *et al*, 2009; Kühner *et al*, 2009; Maier *et al*, 2011, 2013; Chen *et al*, 2016; Trussart *et al*, 2017). Although major progress has been made in dissecting the distinct regulatory layers at the transcriptional level (Yus *et al*, 2019), how regulation occurs at the translational level is still poorly understood. To redress this gap of information, we have analyzed in-depth the impact of regulated proteolysis in the biology of *M. pneumoniae*, which lacks obvious protein adaptors and encodes only two essential ATP-dependent proteases, Lon and FtsH (Himmelreich *et al*, 1996; Lluch-Senar *et al*, 2015). To gain insight into the substrate repertoire of these proteases in *M. pneumoniae*, we used a quantitative proteomics approach combined with RNA expression data to define proteome changes after Lon and/or FtsH depletion. A total minimum of 62 Lon, and 34 FtsH, candidate substrates were identified, of which, some were validated by immunoblotting and *in vivo* degradation assays. We found an enrichment of FtsH substrates associated with the membrane, suggesting a key role of this protease in maintaining membrane protein homeostasis. Supporting this, the cell membrane integrity was found to be compromised after FtsH depletion. Lon candidate substrates were found associated with specific biological pathways, including cell division, DNA repair/recombination, and the restriction-modification system. Furthermore, mutational studies of selected Lon substrates identified specific degrons enriched in hydrophobic residues. Additionally, a genome-wide analysis of protein half-lives revealed an enrichment of short-lived proteins among Lon substrates. Overall, these results show that Lon has important regulatory functions in *M. pneumoniae* and suggest that this minimal organism has evolved to take advantage of the ability of Lon to recognize accessible hydrophobic regions, to regulate precisely the expression of functional native proteins. We also provide evidence that Lon has important roles in protein quality control, degrading misfolded proteins and those not assembled into functional complexes. Finally, by integrating information from proteases and chaperones, we suggest an integrated model of how protein degradation takes place in this bacterium.

## Results

### Protease repertoire of *Mycoplasma pneumoniae*

*Mycoplasma pneumoniae* possesses a small set of intracellular proteases, according to the MEROPS peptidase database (Rawlings *et al*, 2016) and manual annotation (Yus *et al*, 2019). This includes the Lon (MPN332) and FtsH (MPN671) ATP-dependent proteases and homologs to intracellular peptidases involved in peptide degradation, such as proline iminopeptidase Pip (MPN022), oligoendopeptidase F PepF (MPN197), X-Pro aminopeptidase PepP (MPN470), and leucine aminopeptidase PepA (MPN572). Several peptidases and proteases implicated in protein processing and maturation are also present, including the methionine aminopeptidase Map (MPN186), ribosomal-processing cysteine protease Prp (MPN326), and the lipoprotein signal peptidase Lsp (MPN293). The genes of all these proteins except for *pip* (MPN022) are essential for survival based on transposon essentiality studies (Lluch-Senar *et al*, 2015). Surprisingly, no gene encoding a type I signal peptidase (SPase) has been found in the genome, yet there is evidence supporting the presence of SPase-like activity in *M. pneumoniae* (Catrein *et al*, 2005).

Out of all the intracellular peptidases and proteases found in *M. pneumoniae*; Lon is the protease that exhibit larger transcriptional changes and is more commonly affected in its expression upon different perturbations (Yus *et al*, 2019), whereas PepP, Lsp, and PepF are the ones that exhibit less variability in expression, suggesting a housekeeping-like behavior (Fig EV1A and B). Among the perturbations tested, glucose starvation was the condition that negatively disturbed the most the expression level among peptidases and proteases genes, except for *lon* (Fig EV1A). Genes encoding Map, Prp, Lon, and to a lesser extent Pip and FtsH were also induced by cold shock, suggesting that these proteases/peptidases could play a role in cold stress adaptation. Lon and PepA are encoded in operons regulated by the heat-shock transcription factor HrcA (MPN124), while Pip is in an operon with DnaJ (MPN021) that contains a degenerated motif for HrcA. According to this gene organization, all three genes correlate well with transcriptional changes

affecting chaperones regulated by HrcA, including DnaJ (MPN021), DnaK (MPN434), ClpB (MPN531), GroEL (MPN573), and GroES (MPN574) (Fig EV1C).

## Construction of conditional Lon and FtsH mutants in *Mycoplasma pneumoniae*

Lon and FtsH can target folded proteins for unfolding and subsequent digestion via their ATPase unfoldase activity (Sauer & Baker, 2011); in contrast, the other peptidases and proteases present in *M. pneumoniae* can only digest peptides or unfolded proteins. Thus, Lon and FtsH seem to be the main proteases of *M. pneumoniae* that have the capacity to control protein function through protein degradation. Isolation and characterization of null mutants has so far been hampered by the fact that Lon and FtsH are essential for cell growth. To study the cellular functions of these proteases, we overcame these difficulties by generating the first conditional mutants in *M. pneumoniae*. We used genome-editing tools (Piñero-Lambea *et al*, 2020) based on the phage recombinase GP35 (Sun *et al*, 2015) to control Lon and FtsH expression through a Tet-inducible system (see Materials and Methods and Appendix Fig S1). To determine the effect of the absence of both proteases, we also constructed a Lon/FtsH double mutant, by performing the same genome editing within the *lon* locus in the FtsH-inducible mutant.

Lon and FtsH expression monitored by RNA-seq and Western blot assays under inducing and depleting conditions, showed good repression-induction transcriptional pattern, which correlated with protein expression (Fig 1A–C). Compared to the wild-type strain, the inducible system supported Lon and FtsH protein levels slightly below and above, respectively. Proteome-wide measurements of protein half-life revealed protein turnovers for Lon and FtsH of 13 and 52 h (Table EV1). Accordingly, Western blot analysis showed that complete depletion of Lon was not observed until 48 h after removing the inducer from the medium, whereas FtsH was significantly reduced after 72 h of depletion. Quantitative MS analysis indicated a 4-fold ($\log_2$) reduction for both proteases under these depleting conditions (Table EV2). Thus, unless otherwise indicated, depletion experiments for Lon and FtsH were performed at 48 and 72 h, respectively.

## Phenotypic characterization of Lon and FtsH mutants

To gain insights into the cellular functions of Lon and FtsH, we first assessed the effect of depletion of Lon and FtsH in cell growth. Consistent with their reported essentiality, depletion of either of both proteases inhibited growth based on pH (Appendix Fig S2) and cell biomass measurements (DNA and protein) along the growth curve (Fig 2A). Slow down of DNA replication was further confirmed by pulse-chase experiments using the analog bromodeoxyuridine (BrdU) (Appendix Fig S3).

Next, we examined whether Lon or FtsH depletion resulted in increased protein aggregation, as a consequence of possible accumulation of misfolded proteins. For this, we obtained insoluble fractions after Triton X-100 solubilization from mutants grown under inducing or depleting conditions, and we stained them with Thioflavin-T (ThT), a commonly used fluorescent dye to monitor protein aggregation (Morell *et al*, 2008). As shown in Fig 2B, a significant increase in ThT fluorescence intensity was detected after Lon

depletion, which was magnified when Lon mutants were exposed to heat stress conditions. In contrast, no significant differences were observed after FtsH depletion, suggesting that increased protein aggregation is a Lon KO-specific phenotype.

We also explored the roles of Lon and FtsH in the maintenance of the cell membrane using a dye-exclusion assay. We found that depletion of FtsH, but not Lon, compromises the membrane integrity under normal and mildly membrane disruptive conditions (Fig 2 C). In fact, we were unable to regrow FtsH mutant cells following FtsH depletion, suggesting important cellular damage (Fig 2D). This was not the case for the Lon-depleted mutant, which allowed us to assess the role of Lon under proteotoxic stress conditions. Consistent with a role in stress tolerance, Lon-depleted cells exhibited increased sensitivity to heat stress as compared to Lon-expressing cells (Fig 2D).

## Identification of Lon and FtsH candidate substrates

To identify Lon and FtsH candidate substrates in *M. pneumoniae*, we performed label-free quantitative mass spectrometry to compare the proteomes of Lon and FtsH mutants grown in inducing or depleting conditions. A protein was considered as detected if at least one common unique peptide was found in the two biological replicates (see Materials and Methods). Using this criterion, we identified 494–547 proteins, which correspond to a proteome coverage of 67.1–74.3 % with respect to all mutants and conditions analyzed. A total of 73 and 43 proteins were significantly upregulated after Lon or FtsH depletion, respectively, with protein fold changes higher than 2-fold ($\log_2(\text{protein\_FC}) \geq 1$), while 40 and 20 proteins were downregulated after Lon or FtsH depletion, respectively ($\log_2(\text{protein\_FC}) \leq -1$) (Appendix Fig S4).

To define Lon and FtsH candidate substrates, we focused our attention on upregulated proteins, which we classified into differentially detected (DD) or differentially expressed (DE) (see Materials and Methods). As changes in protein levels could result from transcription regulation rather than a decrease in the degradation rate, we integrated RNA-seq data from the protease mutants grown in inducing or depleting conditions into the analysis (Fig 3A and Table EV2). To predict candidate substrates, we applied a cut-off criterion in which we selected DD proteins with fold changes of mRNA levels lower than 2-fold ($\log_2(\text{mRNA\_FC}) \leq 1$) (Appendix Fig S5), and DE proteins with protein/mRNA fold changes higher than 2-fold ($\log_2(\text{protein\_FC}/\text{mRNA\_FC}) \geq 1$). As a result, we obtained a minimum of 62 and 34 candidate substrates for Lon and FtsH, respectively (Table EV3). After applying similar cut-off criteria, downregulation of 19 and 11 proteins could not be explained by decreased mRNA expression of their respective genes upon Lon or FtsH depletion, respectively. Intriguingly, one third of these 30 proteins represented lipoproteins (Table EV2).

Regarding the candidate substrates, only the ortholog of the TsrB glycosyltransferase (MPN028), which has a large cytoplasmic catalytic domain and two C-terminal transmembrane helices, was identified as a common substrate for both Lon and FtsH (Fig 3B), indicating that some intracellular proteins with transmembrane domains can be targeted by both proteases. The low degree of substrate overlapping suggests that Lon and FtsH perform distinct functions in *M. pneumoniae* protein homeostasis, probably due to their different cell location (cytoplasmic for Lon and membrane

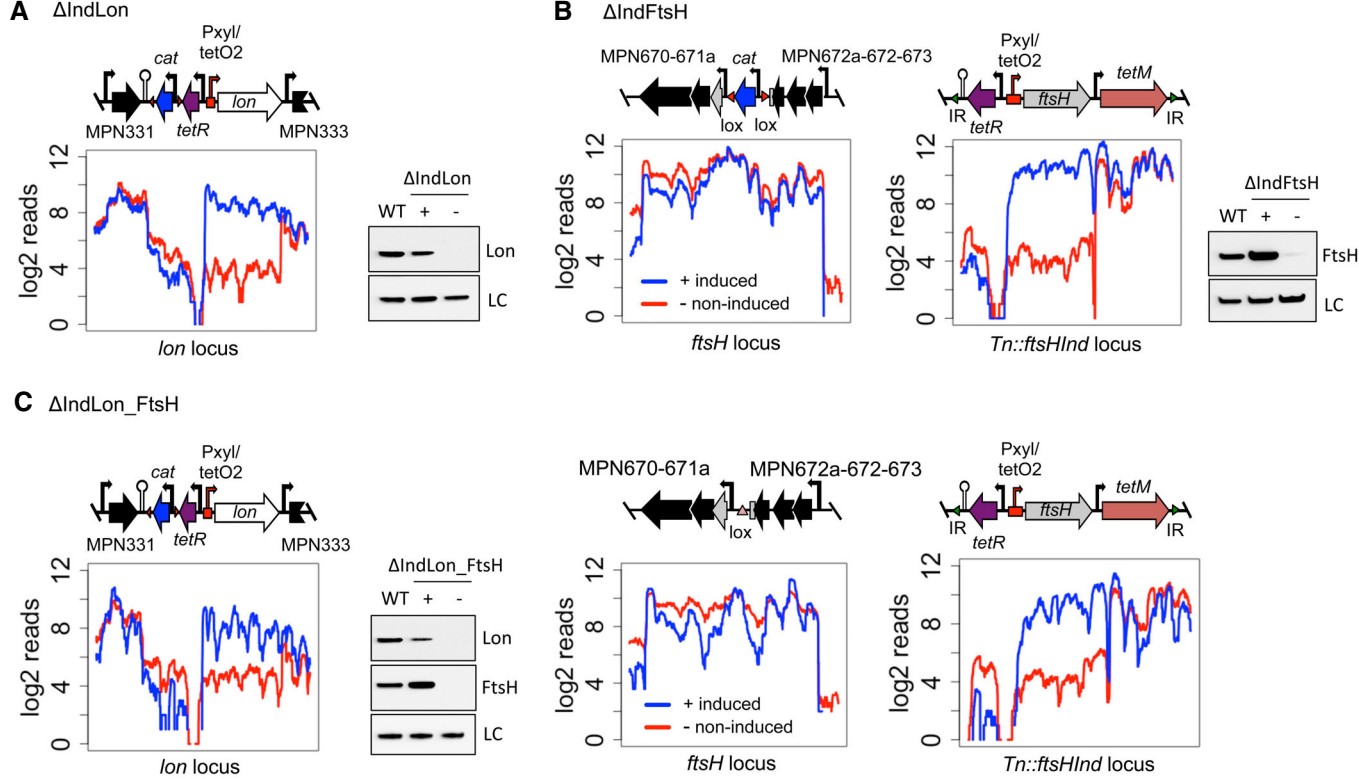

**Figure 1. Construction of Lon and FtsH conditional mutants in *Mycoplasma pneumoniae*.**

A–C  RNA-seq transcriptional profiles across the modified locus, as well as immunoblots assessing protein expression of Lon and FtsH, are shown for ΔIndLon (A), ΔIndFtsH (B), and ΔIndLon_FtsH (C) grown under inducing or depleting conditions (48 and 72 h of depletion for Lon and FtsH, respectively). Symbols +/− indicate inducing or depleting conditions. LC, loading control. WT, wild-type. A schematic representation of the DNA rearrangements in the *lon* and *ftsH* locus is also shown for each strain. The *ftsH*-inducible platform inserted by transposon delivery is shown for ΔIndFtsH and ΔIndLon_FtsH strains. The Pxyl/tetO2-inducible promoter is highlighted with a red bent arrow and the terminator sequence used to isolate the promoter is represented by a hairpin structure. The *tetR* repressor gene and the resistance markers *cat* and *tetM* are indicated in purple, blue, and red, respectively.

anchored for FtsH). Supporting this, 82.3% of the FtsH candidates were membrane-associated proteins, in contrast to 17.7% of the Lon substrates (Fig 3C; half of the Lon membrane-associated substrates have a predicted cytoplasmic domain larger than 25 amino acids). We also determined *M. pneumoniae* protein turnover rates by SILAC-based proteomics (Table EV1). Lon candidate substrates exhibited significant lower protein half-lives as compared to the average [Mann–Whitney–Wilcoxon (MWW) two-sided test, $P = 6.53 \times 10^{-4}$; Fig 3D and Table EV3].

We found that Lon targets were involved in several functional pathways, including cell division, the restriction-modification system, or DNA repair/recombination (Fig 3F). Substrate candidates were found particularly enriched in genes of the functional category "defense mechanisms" (two-sided Fisher test with multiple test correction, family-wise false discovery rate 5%). A putative toxin–antitoxin system, putative transporters, enzymes associated with different metabolic pathways, and proteins of unknown function complete the list. For the FtsH substrates, we identified three components of the Sec secretion pathway including SecD (MPN396), SecY (MPN184), and SecE (MPN068), proteins related to metabolism, some putative transporters, and numerous proteins of unknown function.

**Validation of Lon and FtsH substrates**

Our comparative proteomic analysis suggested that Lon regulates proteins associated with different cellular pathways, including cell division and the restriction-modification system. To further validate the potential substrates involved in these processes, we performed time course depletion experiments and *in vivo* degradation assays on a subset of candidate substrates.

*Mycoplasma pneumoniae ftsA* (*mpn316*) and *ftsZ* (*mpn317*) cell division-related genes are transcribed in a single transcriptional unit together with *mraZ* (*mpn314*) and *mraW* (*mpn315*). Even though these genes are transcriptionally expressed at similar levels, the protein abundance of FtsA and FtsZ is significantly reduced as compared to MraZ and MraW (Fig 4A). While transcript levels remained unaffected, the protein abundances after Lon depletion of FtsA and FtsZ increased significantly, and doubled in the case of MraW. A similar situation was found with the essential gene *mpn525*, which is co-transcribed with *mpn526* and putatively encodes the replication initiation protein DnaB (Fig 4A). To confirm these observations, we expressed FLAG-tagged derivatives of these proteins in the Lon mutant and assessed the protein levels in time course depletion experiments (Fig 4B). In all three cases, the

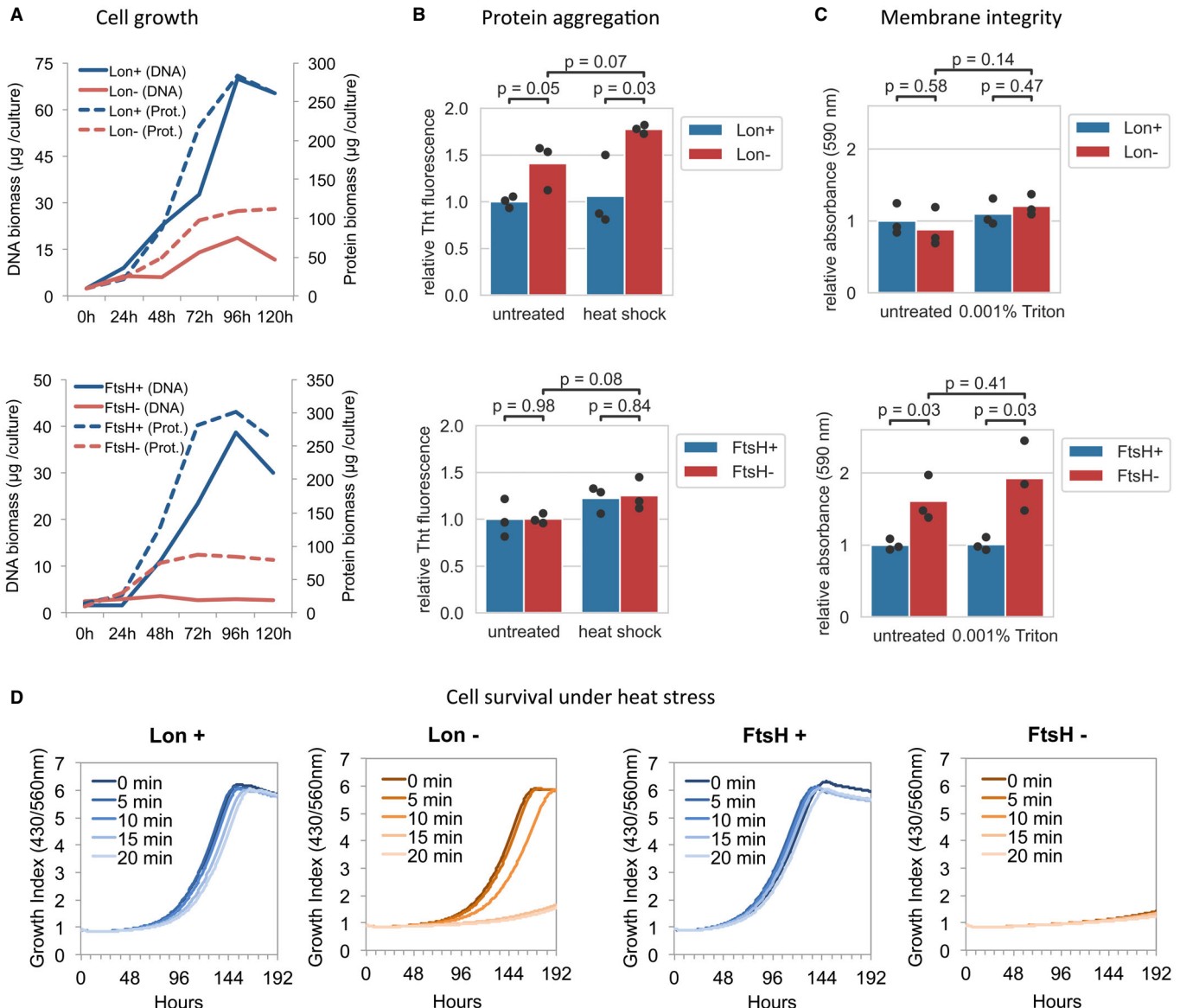

**Figure 2. Phenotypic characterization of Lon and FtsH conditional mutants in *Mycoplasma pneumoniae*.**

A  Cell growth assessment of ΔIndLon (upper plot) and ΔIndFtsH (lower plot) mutants grown under inducing (blue) or depleting conditions (red). Growth was monitored by measuring DNA and protein biomass over time. The average from two independent biological replicates is shown.

B  Protein aggregation in ΔIndLon (upper plot) and ΔIndFtsH (lower plot) mutants grown under inducing (blue) or depleting conditions (red, 48 h, and 72 h of depletion for Lon and FtsH, respectively). Protein aggregates in Triton X-100 insoluble fractions of untreated or heat-shock (15 min at 45°C)-treated cells were measured by Thioflavin (ThT) staining. Bars represent the mean of three biological replicates (dots). Significance of comparisons was assessed by two-sided independent *t*-test (exact *P*-values are shown).

C  Assessment of cell membrane integrity of ΔIndLon (upper plot) and ΔIndFtsH (lower plot) mutants grown under inducing (blue) or depleting conditions (red, 72 h of depletion for both, Lon and FtsH). Membrane integrity of untreated cells or after exposure during 30 min to 0.001% Triton X-100 was assessed by trypan blue exclusion staining. Bars represent the mean of three biological replicates (dots). Significance of comparisons was assessed by two-sided independent *t*-test (exact *P*-values are shown).

D  Role of Lon and FtsH under heat-shock stress conditions. ΔIndLon and ΔIndFtsH mutants were grown under inducing (Lon+ or FtsH+) or depleting conditions (Lon− or FtsH−, 60 h of depletion for both, Lon and FtsH), and then exposed at 45°C during 0, 5, 10, 15, or 20 min. Then, growth after the heat treatment was monitored over time under inducing conditions by the 430/560 absorbance rate index that shows pH changes in the medium. The average from two independent biological replicates is shown for each condition.

expression of the N-terminal FLAG fusion was dependent on Lon expression, indicating these cell division factors are Lon regulated. Unfortunately, we could not perform standard *in vivo* degradation assays to accurately assess the degradation kinetics of these substrates, in part due to the unusual long half-lives of the *M. pneumoniae* proteins (68 h on average based on pulse-chase SILAC

analysis), and the difficulties found inherent to the mutant conditional system used in this study. For example, the majority of the substrates tested are not expressed or barely detected under normal growth conditions; and in contrast to other bacterial systems, overexpression of the substrate to saturate the protease capacity is not feasible in *M. pneumoniae*, which has a low protein capacity

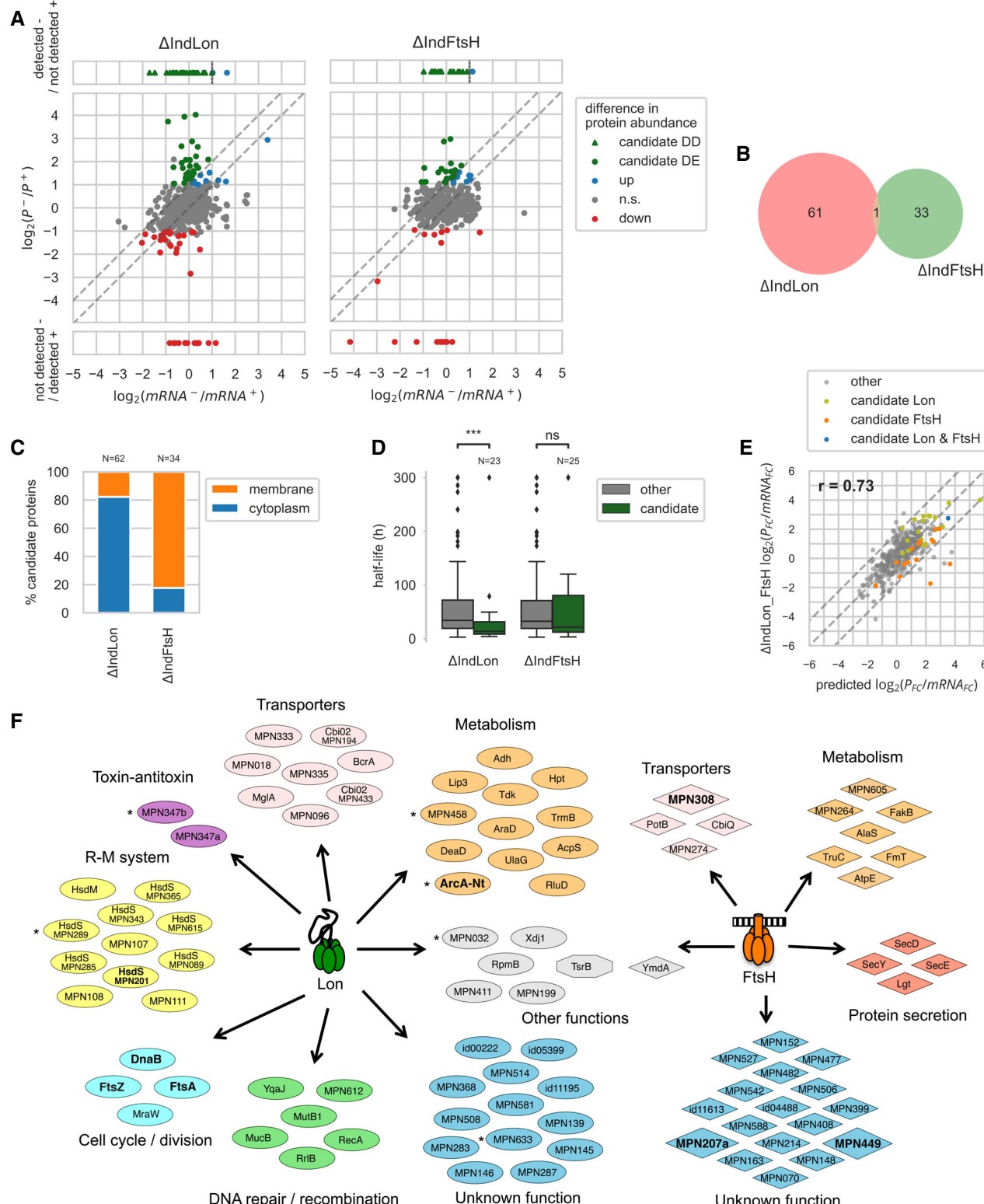

Figure 3.

**Figure 3.   Identification and analysis of Lon and FtsH candidate substrates.**

A   Transcriptional and protein abundance changes between inducing (+) and depleting (−) conditions (48 and 72 h of depletion for Lon and FtsH, respectively) in ΔIndLon and ΔIndFtsH mutant strains. Differentially expressed (DE) Lon/FtsH candidate substrates were identified as proteins with a significant increase in abundance upon protease depletion that could not be attributed to an increase in mRNA levels, i.e. $\log_2(\text{protein\_FC/mRNA\_FC}) \geq 1$ (upper dashed line). Differentially detected (DD) candidates were identified as proteins not detected in the induced condition and detected in the depleted condition, and whose mRNA level did not increase more than 2-fold (vertical dashed line).

B   Overlap of substrate candidates between the two mutant strains.

C   Proportion of membrane and cytoplasmic proteins among candidate substrates.

D   Distribution of half-lives measured by a SILAC time course experiment (average from 2 to 4 biological replicates) for Lon/FtsH candidate substrates as compared to the other proteins. Lon candidate substrates showed significantly shorter half-lives (MWW two-sided test, ***$P = 6.53 \times 10^{-4}$). Box shows the quartiles of the distribution, line shows the median, whiskers extend to 1.5 times the inter-quartile range past the low and high quartiles and define the limits for outliers, shown as points.

E   Changes in the protein to mRNA fold changes ratio, $\log_2(\text{protein\_FC/mRNA\_FC})$, in the double mutant strain ΔIndLon_FtsH as compared to the predicted ratio, computed as the sum of the $\log_2$ ratios of both individual mutants. Proteins that showed a significant change in protein level in at least one of the individual mutants are shown (374 proteins). Proteins whose ratio deviated significantly from the predicted one (upper and lower dashed diagonal lines) reveal epistasis effects.

F   Functional classification of candidate substrates. Validated substrates are highlighted in bold. Asterisks indicate candidate substrates classified as pseudogenes or truncated gene variants.

production. As an alternative method, we first depleted the protease to stabilize the substrate, and then, we transiently induced the protease before blocking protein synthesis. Substrate expression was then compared with non-induced cells at different time points after antibiotic treatment, showing that FtsA, FtsZ, and DnaB-like protein were degraded more slowly under Lon depleting conditions (Fig 4C). Drawbacks of this alternative method are that cells need some time to recover after depletion and that the accumulation of other substrates or misfolded proteins during protease depletion may saturate the protease capacity. Since these circumstances can interfere with the degradation kinetics, this alternative method can only be considered as a qualitative degradation assay.

Another interesting group of proteins whose expression was upregulated in the absence of Lon was associated with the restriction-modification system of *M. pneumoniae*. Except for the HsdS subunit encoded by *mpn638*, which is highly expressed, the other eight HsdS protein subunits present in *M. pneumoniae* are not expressed or are barely detected (Maier *et al*, 2011; Lluch-Senar *et al*, 2015; Miravet-Verde *et al*, 2019). The same is true for the several putative methylases (except MPN342) or restriction enzymes encoded in the genome. With the exception of the HsdS subunits encoded by *mpn638* and *mpn507*, all HsdS and putative methylases were found to be significantly upregulated after Lon depletion (Fig 4 D). Although MPN507 did not meet our established cut-off criteria, we also observed a moderate increase for this HsdS subunit. In contrast, we did not find changes in the levels of the putative restriction enzymes, which are expressed at very low level and probably represent pseudogenes, since they contain frameshift mutations that split these genes in two (*mpn109–110*) and three fragments (*mpn345–347*) (Table EV4). To verify these results, we constructed N- or C-terminal FLAG fusions of MPN201 as a representative of an unstable HsdS, and MPN638 as a stable HsdS subunit. In agreement with MS data, we found that the HsdS subunit MPN201 was only detected in the absence of Lon, whereas as expected the levels of the HsdS subunit MPN638 were not influenced by the expression of Lon (Appendix Fig S6). Additional time course depletion experiments confirmed that the stability of the HsdS subunit MPN201 is dependent on Lon expression (Fig 4E). In the case of the *in vivo* degradation assay (Fig 4F), we could not observe a significant decrease in HsdS (MPN201), suggesting either that it is indirectly

regulated or the conditions used were not sensitive enough to reveal an effect.

Finally, our proteomic analysis showed that membrane proteins are the main targets affected by *M. pneumoniae* FtsH. To validate these results, we constructed N- and C-terminal FLAG variants of six candidates that showed significant changes between the induced and the FtsH-depleted strains (Fig 4G and Appendix Fig S7). These included a lipoprotein (MPN152), an small ORF [MPN207a; (Miravet-Verde *et al*, 2019)], Lgt (MPN224), a putative permease (MPN308), an integral membrane protein (MPN449), and a protein of unknown function (MPN527). Out of these 6, we did not detect expression or it was very weak for MPN152, Lgt (MPN224), and MPN527 (Appendix Fig S7). For the other three candidates, we confirmed by time course depletion experiments that their expression level was dependent on FtsH activity (Fig 4H). These results were supported by qualitative *in vivo* degradation assays, which showed delayed substrate degradation in the absence of FtsH (Fig 4I).

### Identification of protease recognition determinants

Global sequence analyses of the candidate substrates did not reveal clear common motifs that could define a Lon or FtsH degradation signal. As accessibility of a degron may be a key factor in determining degradation, we focused our attention on the N- and C-terminal tails, as these regions should be generally more accessible. We searched for a statistical enrichment of specific physicochemical properties in these domains (Appendix Fig S8). However, we did not find a clear pattern, possibly due to insufficient statistical power or to the presence of confounding factors.

As an alternative strategy, we focused our attention on the analysis of specific substrates. In contrast to the N-terminal fusions, we found that FLAG fusions at the C-terminus of FtsA, FtsZ, and DnaB-like protein promoted protein stability even in the presence of Lon (Appendix Fig S6). These observations suggested that Lon degrons in these proteins may lie close to the C-terminus, and they could be masked by the presence of the tag. Examination of the C-terminal sequences of these proteins revealed several motifs enriched in hydrophobic residues (Fig 5A), which have been previously associated with Lon-dependent degradation (Gur & Sauer, 2008). Subsequent mutational analysis of the hydrophobic residues of these

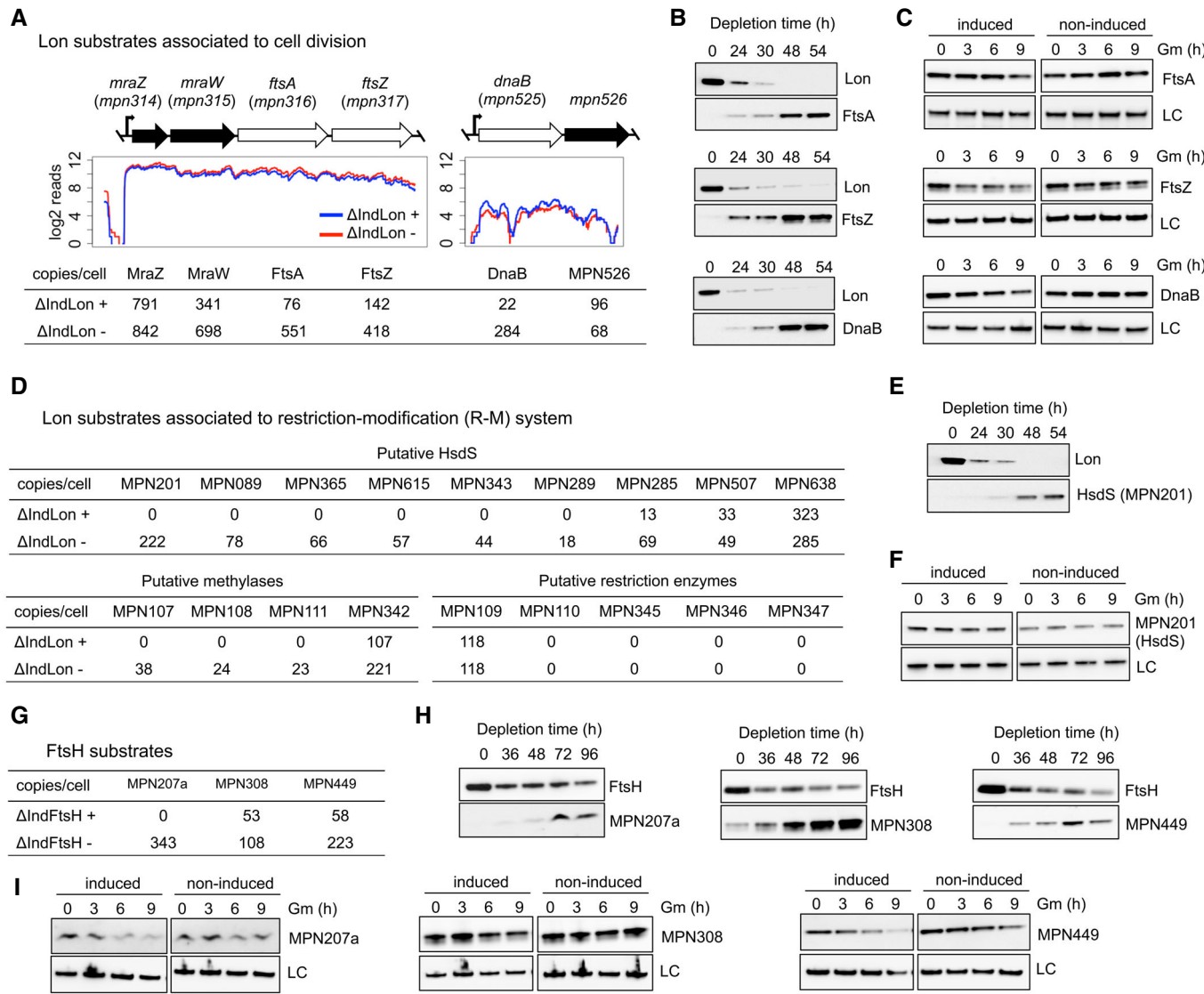

**Figure 4. Validation of selected Lon and FtsH candidate substrates.**

A Transcription profiles of ΔIndLon mutants grown in inducing (+) or depleting (−) conditions across the cell division and the MPN525 (DnaB) operons. The respective estimated protein copy numbers based on MS data are shown below.

B Time course experiments showing depletion levels of Lon over time and the corresponding accumulation of N-terminal FLAG-tagged derivatives of candidate substrates associated with cell division, including FtsA, FtsZ, and DnaB. Protein levels of Lon and candidate substrates were assessed by immunoblot analysis using anti-Lon and anti-FLAG antibodies. LC, loading control.

C *In vivo* degradation assays of FtsA, FtsZ, and DnaB. ΔIndLon mutants expressing the different N-terminal FLAG-tagged derivatives were grown in depleting conditions for 36 h. Then, Lon expression was transiently induced for 3 h before translation was blocked with gentamicin (Gm). Samples were taken at the indicated time points after gentamicin treatment, and processed for immunoblot analysis using anti-FLAG antibodies. LC, loading control. As controls, non-induced cells were also treated with gentamicin and processed at the same time points. Immunoblots are representative of two independent experiments.

D Estimated protein copy numbers based on MS data of components associated with the R-M system.

E, F Similar to panel B (E) and C (F) but for a C-terminal FLAG-tagged derivative of MPN201 as a representative of an unstable HsdS subunit.

G Estimated protein copy numbers based on MS data of FtsH candidate substrates.

H Similar to panel B but for C-terminal FLAG-tagged derivatives of FtsH candidate substrates. Protein levels of FtsH and candidate substrates were assessed by immunoblot analysis using anti-FtsH and anti-FLAG antibodies. LC, loading control.

I Similar to panel C, but for FtsH candidate substrates. In this case, C-terminal FLAG-tagged derivatives were grown in depleting conditions for 60 h before transient expression of FtsH for 3 h and gentamicin treatment.

motifs demonstrated that LVQKLI and FNY, which were located at the very end of the C-terminus of FtsA and FtsZ, respectively, were required for Lon recognition and degradation (Fig 5B). In a similar manner, we also found that mutations in LGLKYV and VNYFL motifs, present in the C-terminal region of FtsZ and DnaB-like, resulted in protein stabilization in the presence of Lon (Fig 5B).

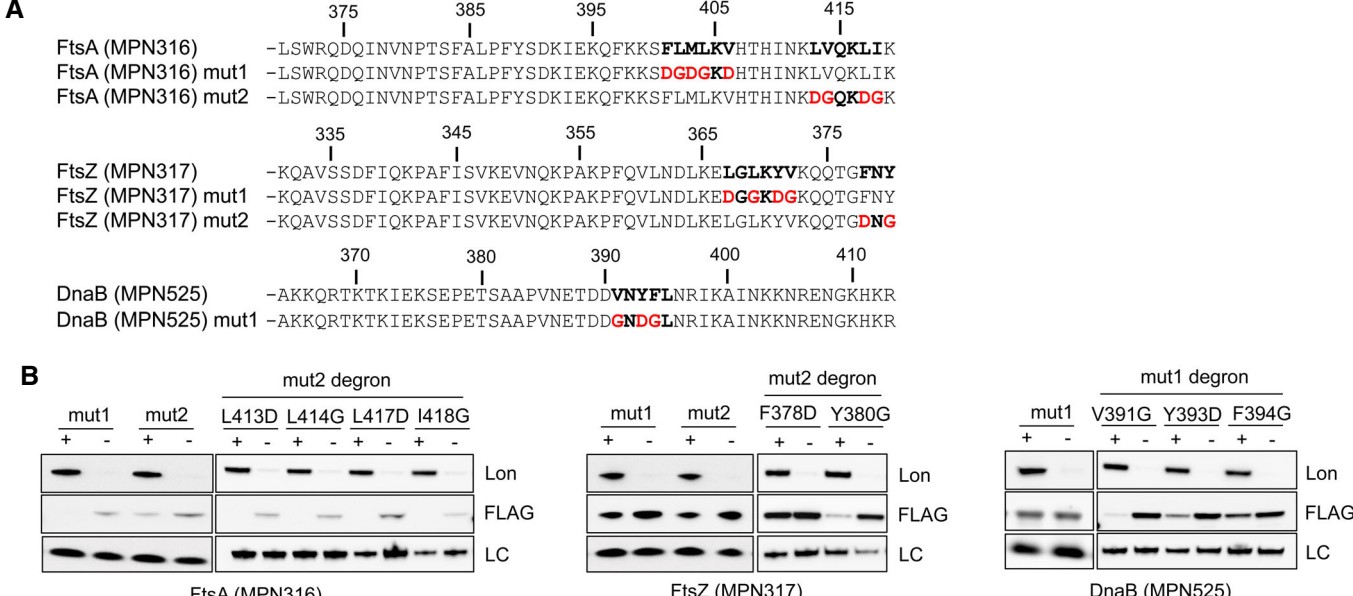

**Figure 5. Mutational analysis to identify Lon degrons in Lon substrates.**

A   Amino acid sequence of the last 50 residues of FtsA (MPN316), FtsZ (MPN317), and DnaB (MPN525) proteins and their mutant derivatives. Hydrophobic motifs that can potentially act as Lon degrons are highlighted in bold. Mutations performed in these putative Lon degrons are shown in red.

B   Protein stability assessment of Lon substrates containing multiple or single mutations in putative Lon degrons. N-terminal FLAG-tagged derivatives with mutations shown in panel A were expressed in the ΔIndLon mutant. Protein levels were then determined by immunoblot using anti-Lon and anti-FLAG antibodies comparing inducing (+) or depleting (−) conditions (48 h of depletion). LC, loading control.

When single amino acid substitutions of these degrons were analyzed, we found that in the case of the LVQKLI degron, none of the individual mutations protected FtsA from degradation, suggesting that several residues within this motif are critical for recognition. In contrast, single substitutions in residues F378 in FtsZ and Y393 or F394 in DnaB-like were sufficient to significantly stabilize these proteins. Overall, these results suggest that the presence of a small hydrophobic region, located in an accessible area, such as a free C-terminal tail, can be sufficient to trigger Lon-mediated degradation in *M. pneumoniae*. We also scanned for similar Lon degradation signals in the HsdS subunit MPN201 and the ArcA pseudogene MPN304, but mutational analysis failed to identify a clear degradation motif, suggesting the existence of multiple exposed degrons in these proteins (Appendix Figs S9–S11).

**Cellular response to Lon and FtsH depletion**

Consistent with the low degree of substrate overlapping, correlation analyses of RNA-seq data revealed different transcriptional responses to the absence of Lon or FtsH (Pearson correlation coefficient, $r = 0.24$; Fig EV2). Similarly, global changes in protein levels in the two individual mutants were largely independent ($r = 0.01$; Appendix Fig S12). To examine how these two responses interact upon simultaneous depletion of both proteases, we compared the $\log_2$ ratios of protein/mRNA fold changes in the double mutant to the sum of the $\log_2$ ratios in the two individual mutants (Fig 3E). Interestingly, the fold change ratios in the double mutant followed the predicted fold changes (within noise) for the vast majority of the proteins (95.4%). This overall absence of epistasis corroborates that

separate responses combine in an independent manner following the simultaneous depletion of Lon and FtsH.

To further examine the cellular response upon Lon and FtsH depletion, we analyzed changes in expression of known transcription factors and indirect regulators associated with *M. pneumoniae* gene regulation (Yus *et al*, 2019). Except RecA (MPN490) and the hypothetical lipoprotein MPN506, none of these known regulators were classified as targets of Lon or FtsH degradation. However, the WhiA-like repressor (MPN241), which has been associated with repression of the main ribosomal protein operon in *M. pneumoniae* (Yus *et al*, 2019), and the membrane-anchored PrkC kinase (MPN248) were moderately upregulated (0.78 and 0.98 $\log_2$, respectively) at the protein level after Lon or FtsH depletion, respectively. We observed that ribosomal proteins represent half of the proteins downregulated in the absence of Lon (16 of 40), a phenotype not associated with FtsH depletion and explained by a reduction in transcript levels (Fig EV2G). Thus, upregulation of WhiA could explain the decrease in ribosomal proteins under Lon depleting conditions. We also found that mRNA and protein levels (−0.8 and −1.1 $\log_2$, respectively) of the Fur repressor (MPN329) were downregulated after Lon depletion, perhaps as a response of protein upregulation of two Lon candidates substrates encoding metal transporters (MPN194 and MPN433). Other transcriptional changes observed seem to be mediated by non-canonical factors or regulation of the activity of transcriptional regulators. For example, depletion of Lon but not of FtsH led to significant increases in genes regulated by the transcription factor HrcA, including *dnaK*, *clpB*, *dnaJ*, *groEL/ES*, and the operon-associated proteases *pepA* and *pip* (Table EV2). These results suggest that the GroEL/ES chaperonin system becomes

saturated in the absence of Lon due to increased protein misfolding and aggregation, leading to the inactivation of the HrcA repressor activity (Mogk *et al*, 1997) and the subsequent transcriptional activation of HrcA-regulated genes. This response could be interpreted as an attempt to deal with protein quality control functions in the absence of Lon by increasing chaperone activity.

### Role of Lon as a quality control protease in *M. pneumoniae*

Lon plays a key role in maintaining proteome homeostasis by degrading damaged and misfolded proteins in other organisms. To assess this protein quality control function in *M. pneumoniae*, we generated an unstable variant of the luciferase reporter using the protein design software FoldX (Schymkowitz *et al*, 2005; Delgado *et al*, 2019) (see Materials and Methods), and its expression was evaluated in the presence or absence of Lon. Consistent with proteolytic activity on unstructured and misfolded proteins, we observed that the designed unstable variant (Fluc_F89E) was more stable in the absence of Lon (Fig 6A). We also analyzed protein expression of genes that are apparently truncated in the genome. We reasoned that some of these gene variants may produce unstable proteins, which could be targeted by Lon. Out of the 36 putative pseudogenes detected in the genome (Table EV4), 23 were not detected by MS in any of the conditions analyzed, suggesting the absence of efficient translation initiation signals and/or low transcription levels. In fact, nine of these pseudogenes exhibited transcript levels below the 10[th] quantile. Among the truncated gene variants with transcript levels above this threshold, we found that a significant proportion were upregulated at the protein level in the absence of Lon (two-sided Fisher test, $P < 0.020$, see Materials and Methods). Six of them were classified as Lon substrates, including MPN633 (unknown function), MPN458 (truncated putative OppA variant), MPN347b (N-terminal fragment of a putative toxin from a toxin–antitoxin pair), MPN304 (N-terminal region of ArcA pseudogene), MPN289 (truncated HsdS), and MPN032, an orthologue of MPN294 (probably an oxidative stress chaperone) containing a premature stop codon (Fig 3F). We also found five gene variants that were upregulated in the absence of Lon, yet they did not meet our established cut-off criteria to be classified as substrates, and therefore, they were not included in our statistical enrichment analysis. One particular example is the non-functional arginine deiminase pathway encoded by *M. pneumoniae* (Rechnitzer *et al*, 2013). This pathway is truncated because of frameshift mutations that have split *arcA* into two pseudogenes, namely *mpn304* and *mpn305*, which encode the N- and C-terminal regions of ArcA, respectively. In addition, *mpn306* that encodes ArcB lacks the first 70 N-terminal amino acids. Only *mpn307* that encodes ArcC and *mpn560* which also shows similarity with *arcA* are complete, although *mpn560* seems to encode an inactive enzyme (Rechnitzer *et al*, 2013). In standard growth conditions, the components of this operon are undetectable or expressed at low protein levels, except for the full length ArcA product (MPN560) (Fig 6B). MS analysis revealed that MPN304 (N-terminal region of ArcA pseudogene) is especially upregulated in the absence of Lon, an observation confirmed by time course depletion experiments and *in vivo* degradation assays using FLAG-tagged derivatives (Fig 6B). Although MPN305 (C-terminal region of ArcA pseudogene) did not meet our established cut-off criteria to be classified as a substrate, we also observed a moderate increase for this pseudogene (Fig 6B).

Overall, these results suggest that truncated variants and/or misfolded proteins are targets of Lon surveillance.

Our initial analysis also revealed increased protein aggregation in the absence of Lon (Fig 2B). To examine whether Lon substrates tend to aggregate when accumulated, we performed MS analysis of the insoluble fractions under inducing and depleting conditions. Compared to the whole cell lysate, we found an enrichment of Lon substrates in the insoluble fractions from Lon-depleted cells (Fig 6 C). This enrichment was not observed in the presence of Lon, although several substrates are barely detected in this condition (Appendix Fig S13). As expected, terminal organelle-associated proteins known to be present in the Triton X-100 insoluble fraction (Regula *et al*, 2001) were found enriched in both Lon conditions. Additionally, immunoblot analysis of soluble and insoluble fractions after Triton X-100 fractionation of wild-type cells showed that Lon remains mainly in the soluble fraction similar to the RL-7 ribosomal protein and opposite to the HMW1 insoluble protein controls (Fig 6 D). In the case of Lon, however, we observed a shift into the insoluble fraction after heat stress conditions, suggesting association of Lon with protein aggregates.

Finally, protein quality control can also be important for maintaining the proper stoichiometry in protein complexes. *Mycoplasma pneumoniae* spends large amounts of energy to express and assemble numerous proteins in a hierarchical manner to form the terminal organelle, a complex cytoskeleton-like structure promoting cell attachment and motility (Krause, 2001). How the assembly of the terminal organelle is regulated is unclear, but maintaining the proper stoichiometry of its components is critical. For example, loss of HMW2 (MPN310) results in instability of several terminal organelle proteins, including HMW1 (MPN447), P65 (MPN309), and P30 (MPN453) (Krause, 2001). To test whether Lon mediates this specific protein turnover, we deleted the N-terminal region of *hmw2* through GP35 recombination in the Lon mutant background (Fig 6E). As previously observed, loss of HMW2 resulted in reduced levels of P65 and P30 in the presence of Lon, but the levels of these proteins were recovered following Lon depletion, indicating that Lon is responsible for this protein turnover (Fig 6E). Importantly, this phenotype was only observed in the absence of HMW2, suggesting that Lon plays a role in adjusting protein levels of subunits that form complexes by degrading those that are not incorporated.

### Global model for protein folding and degradation in *M. pneumoniae*

Based on the above results, and taking into account information previously published on the functional assignment of proteins as well as on transcriptional regulation in *M. pneumoniae* (Yus *et al*, 2019), we propose a comprehensive integrated qualitative model about protein folding, degradation, and regulation in this genome-reduced bacterium (see Fig 7). We show that despite having a reduced set of proteases and an apparent lack of factors providing substrate specificity, *M. pneumoniae* is capable of regulating different cellular processes in a Lon-dependent manner (Fig 7, left panel). Essentially, we propose that, rather than the activity of the protease being regulated per se, regulation occurs through the formation of protein complexes that hide Lon degrons and thus determine the levels of the target proteins. For example, formation

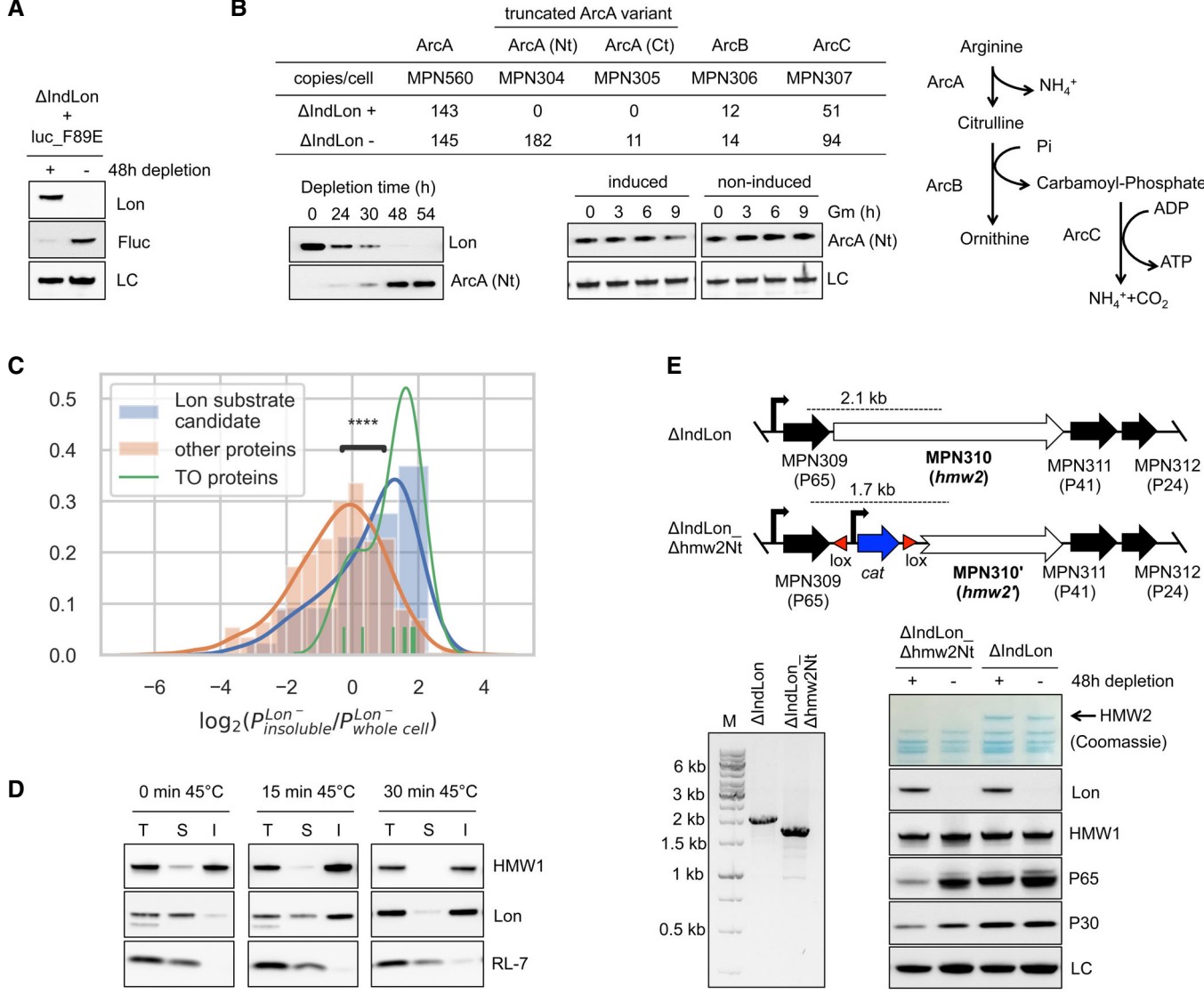

**Figure 6. Role of Lon as a protein quality control protease.**

A  Protein stability of an unstable firefly luciferase variant (Fluc_F89E) assessed by anti-Fluc antibodies in ΔIndLon mutants grown in inducing (+) or depleting (−) conditions (48 h of depletion). LC, loading control.

B  Lon surveillance of truncated protein variants. As an example, an N-terminal FLAG-tagged derivative of MPN304, which encodes a truncated variant of ArcA (ArcA-Nt), was expressed in the ΔIndLon mutant, and its accumulation monitored over time under non-inducing conditions using anti-FLAG antibodies. Anti-Lon antibodies were used to monitor Lon depletion along the time course experiment (Western blot figure on left). MPN304 stability was also assessed by *in vivo* degradation experiments in Lon-depleted mutants (36 h of depletion), in which protein expression was blocked with gentamicin (Gm) during the indicated time points after transient induction of Lon for 3 h. As control, samples after Gm treatment were also taken from non-induced samples. LC, loading control (Western blot figure on right). Immunoblots are representative of two independent experiments. A schematic representation of the arginine deiminase pathway, which is inactive in *M. pneumoniae*, and the estimated protein copy number of its components are also shown for the ΔIndLon mutant grown in inducing (+) or depleting (−) conditions.

C  MS analysis showing the distribution of $log_2$ fold changes in the Triton X-100 insoluble fraction compared with the whole cell lysate, under Lon depleting conditions. Proteins associated with the terminal organelle (TO proteins), known to be enriched in the insoluble fractions, are shown as reference. Lon substrate candidates exhibit larger $log_2$ fold changes compared with the other proteins (Mann–Whitney–Wilcoxon two-sided test, ****$P = 0.0003$), showing enrichment in the insoluble fraction under Lon depleting conditions.

D  Lon enrichment in protein insoluble fractions during heat stress conditions. Lon levels in total (T) and Triton X-100 soluble (S) and insoluble (I) fractions were assessed by immunoblot analyses in untreated wild-type cells or exposed at 45°C during the indicated time points. As controls, the Triton X-100 insoluble protein HMW1 and the soluble ribosomal protein RL-7 were also analyzed by using antibodies raised against these proteins.

E  Protein quality control of terminal organelle assembly mediated by Lon. Top of panel E, a schematic representation of the genetic organization at the P65 operon after deleting the N-terminal region of the *hmw2* gene in the ΔIndLon mutant. The expected PCR products for the deletion are also indicated. Below, an agarose electrophoresis gel showing the PCR validation of the intended deletion, and SDS–PAGE and immunoblot analyses monitoring the expression of specific terminal organelle proteins, whose stability depend on HMW2, for ΔIndLon or ΔIndLon_Δhmw2Nt cell lysates grown in inducing (+) or depleting (−) conditions (48 h of depletion). LC, loading control.

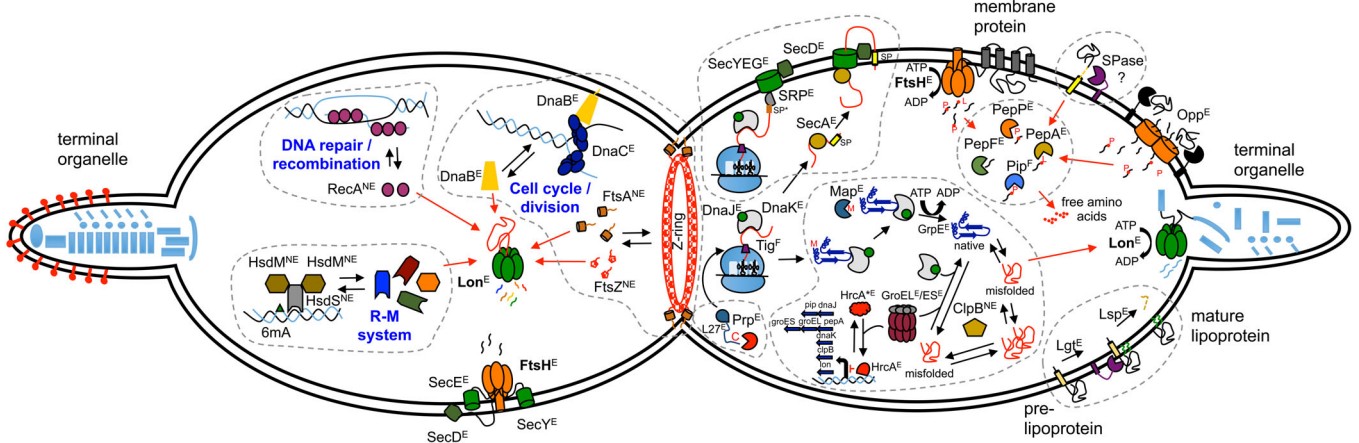

**Figure 7. Integrative model of protein degradation and homeostasis in *Mycoplasma pneumoniae*.**

Schematic view of a *M. pneumoniae* cell during cell division. The right cell compartment shows how protein homeostasis and quality control occur in a minimal cell organism. Protein folding processes with the main chaperones involved are shown: MPN021 (DnaJ), MPN120 (GrpE), MPN331 (Tig), MPN434 (DnaK), MPN531 (ClpB), MPN573 (GroEL), and MPN574 (GroES). Translocation of proteins across the cell membrane through the Sec-translocation pathway is also shown, including the role of FtsH in regulating its proper assembly. Components of the Sec pathway are MPN184 (SecY), MPN068 (SecE), and MPN396 (SecD). Secreted and membrane-associated proteins containing signal peptides are delivered to the membrane by MPN210 (SecA) or co-translationally through the signal recognition particle MPN061 (SRP). All intracellular proteases and peptidases encoded in *M. pneumoniae* are also represented with their main functions. These include MPN671 (FtsH) and MPN332 (Lon) ATP-dependent proteases, and peptidases involved in peptide degradation such as MPN022 (Pip), MPN197 (PepF), MPN470 (PepP), MPN572 (PepA), and in protein processing and maturation, such as MPN186 (Map), MPN326 (Prp), and MPN293 (Lsp). Regulation of specific protease and chaperone genes through the MPN124 (HrcA) regulatory system is illustrated. Also the presence of a putative SPase-I like protein (not yet identified) and the Opp transport system is shown. The left cell compartment shows the major regulatory functions of Lon- and FtsH-controlling specific cellular pathways. Finally, essentiality of the genes involved in this model is shown as E, essential; F, fitness; NE, non-essential.

of the Z-ring would hide the C-terminal degrons of FtsZ and FtsA. Also, we demonstrate that Lon can degrade unassembled components of the attachment organelle. We propose that FtsH may have a similar role in the control of the assembly of the Sec protein translocation complex.

We further show the role of the different protein components that determine the life of a protein in *M. pneumoniae* (Fig 7, right panel). Transposon essentiality studies indicate that all proteins involved are essential for cell survival under normal growth conditions, except for the trigger factor (Tig, MPN331), Pip and the ClpB chaperone. Despite *tig* and *pip* encoding genes are non-essential, they are classified as fitness genes, meaning that transposon insertions in these genes negatively impact but do not prevent growth (Lluch-Senar *et al*, 2015). *Mycoplasma pneumoniae* keeps the bare minimum number of components to ensure correct folding (GrpE, DnaJ, DnaK, GroEL/ES) and export of its membrane proteins (SRP, SecA, SecDEY complex). Regarding protein degradation, Lon and FtsH ensure the unfolding and processing into peptides of misfolded or unassembled proteins, while PepF, PepP, and PepA ensure degradation of these peptides and those imported by the Opp transporter to individual amino acids. Additional proteins include an essential specific protease that processes the L27 ribosomal protein, and the lipoprotein peptidase and acylation system. Of all these components, only the chaperones DnaJ, DnaK, ClpB, and GroEL/ES, and the proteases Pip, PepA, and Lon, have specific transcriptional regulation by the heat-shock regulator HrcA. This is a beautiful example of autoregulation where stress conditions that lead to protein misfolding or aggregation result in saturation of GroEL/ES and consequently misfolding of HrcA, thus liberating the repressed

promoters for these genes (Mogk *et al*, 1997). In fact, this is what we observe after Lon depletion, corroborating the role of Lon as a quality control protease that degrades misfolded proteins. ClpB would not be essential under normal growth conditions, since its role would be mostly to untangle aggregated proteins that could be present under stress conditions. Finally, the fact that we do not see an increase in protein expression of the HrcA-regulated genes after FtsH depletion suggests that membrane proteins do not depend on these chaperones.

## Discussion

Lon and FtsH are essential genes in *M. pneumoniae*, indicating that both proteases are crucial in maintaining proteome homeostasis and/or in regulating key cellular processes. In more complex bacteria, the essentiality of Lon and FtsH is not always seen (Deuerling *et al*, 1997; Ogura *et al*, 1999; Ruvolo *et al*, 2006), probably indicating functional redundancy among the proteases encoded by their genomes. To cope with Lon and FtsH essentiality in *M. pneumoniae*, we conditionally depleted both proteases and analyzed proteomic and transcriptional changes.

The integration of RNA-seq data into our proteomic study allowed us to predict the proteome changes that were most likely due to Lon- or FtsH-mediated degradation, and not to indirect transcriptional changes induced by other factors affected by these proteases. Under our strict criteria, we found a minimum of 62 and 34 candidate substrates for Lon and FtsH, respectively. With the exception of MPN028, we did not find common substrates,

indicating that they have two clearly separate functions, probably determined by the different localization of the two proteases. Accordingly, depletion of Lon and FtsH was accompanied by different transcriptional responses that combine in the double mutant. Consistent with the membrane localization of FtsH, we found that depletion of FtsH affects the integrity of the cell membrane, suggesting a critical role for this protease in maintaining membrane protein homeostasis. In agreement with this, the majority of FtsH substrates identified were membrane-associated proteins (i.e., SecY, SecD, and SecE of the Sec protein translocation pathway). SecY and SecD were previously identified as FtsH substrates in *E. coli* (Kihara *et al*, 1995; Arends *et al*, 2016): In contrast, SecE seems to be a poor substrate in this species (Chiba *et al*, 2002). SecY is degraded when it fails to assemble with its interacting partner SecE. It has been proposed that the short N-terminal tail of SecE may limit efficient FtsH recognition in *E. coli* (Chiba *et al*, 2000). In fact, the *E. coli* SecE has three spanning membrane regions, whereas its *M. pneumoniae* counterpart is predicted to have a single transmembrane segment with a long N-terminal tail facing the cytoplasm. This topological difference could explain why, unlike in *E. coli*, SecE is efficiently degraded in *M. pneumoniae* when not complexed to SecY. This interpretation favors the idea that the length of the cytosolic tails is also important to initiate FtsH-mediated degradation in *M. pneumoniae* as found in *E. coli* (Chiba *et al*, 2000, 2002).

Similar to FtsH, Lon plays a major role as a protein quality control protease in other organisms. In this regard, we showed that depletion of Lon results in increased protein aggregation, suggesting accumulation of misfolded proteins in the absence of this protease. Indeed, we detected both a significant enrichment of Lon substrates in insoluble fractions, and the accumulation of Lon in these fractions under proteotoxic stress conditions. In agreement with these observations, Lon-depleted cells were more sensitive to heat stress, consistent with a role of Lon in stress tolerance. Lon substrates also exhibited statistically shorter half-lives as compared to the average, suggesting a role for this protease in modulating general protein decay. Similarly, a significant number of proteins that are generally not detected in normal growth conditions become detectable after depletion of Lon, without an increase in RNA expression. This finding can be interpreted either as specific protein regulation, or as protein quality control of unstable, non-functional proteins encoded by genes that accumulated frameshift or point mutations through evolution, as we have shown with an engineered unstable variant of the luciferase reporter. We also demonstrated that Lon is involved in the turnover of proteins that fail to assemble correctly in the terminal organelle. In fact, Lon and several chaperones localize to this complex structure (Nakane *et al*, 2015), suggesting an active role of these quality control proteins in regulating the proper assembly of the terminal organelle.

Our results suggest that the essential character of Lon and FtsH in *M. pneumoniae* is due to their involvement in maintaining proteome homeostasis, yet the possibility that the stabilization of a specific substrate could lead to a lethal phenotype cannot be discarded. For instance, the essentiality of FtsH in *E. coli* seems to be associated with its role in regulating LpxC (Ogura *et al*, 1999). To identify possible suppressor mutations, we performed a transposon mutagenesis analysis in the Lon and FtsH mutant strains (see Appendix Fig S14). However, we were unable to isolate a mutant,

in which the inducible system was repressed, suggesting that stabilization of a single non-essential protein is unlikely to explain the lethal phenotype. Additionally, the expression of substrates (FtsA, FtsZ and DnaB-like) that could be stabilized in the presence of Lon did not result in cell death.

Our study also indicates that Lon is involved in regulation of diverse but specific cellular pathways of *M. pneumoniae*. For example, proteins related to recombination and DNA repair were identified as Lon candidate substrates, including RecA (MPN490), RrlB sigma accessory protein [MPN534, (Torres-Puig *et al*, 2018)], and DNA polymerase IV (MPN537). Together with ClpXP, Lon also regulates the levels of the error-prone polymerase UmuD protein in *E. coli* (Frank *et al*, 1996) and is part of the SOS response of *Pseudomonas aeruginosa* by indirectly modulating RecA function (Breidenstein *et al*, 2012). Thus, it seems that there is a common role for Lon to cope with different cellular stresses. *Mycoplasma pneumoniae* apparently lacks an SOS system and phase and antigenic variation seem to be the major outcomes of recombination activity in *M. genitalium*, the closest relative to *M. pneumoniae* (Burgos *et al*, 2012). These observations suggest that Lon may be part of the underlying regulatory mechanism triggering this immune recombination evasion system (Burgos *et al*, 2012; Burgos & Totten, 2014).

Our data also strongly support the involvement of Lon in cell division of *M. pneumoniae*. We found that both FtsZ and FtsA cell division factors (and probably MraW) are substrates of Lon in *M. pneumoniae*. ClpXP likewise modulates cell division through degradation of FtsZ in *E. coli* and *C. crescentus* (Camberg *et al*, 2011; Williams *et al*, 2014). Therefore, it seems that Lon has substituted ClpXP in order to regulate Z-ring formation and disassembly in *M. pneumoniae*. MPN525 (DnaB-like) is also a Lon substrate. Although the exact function of MPN525 has not been determined experimentally, sequence homology analysis suggests it may function as a helicase loader analogously to the replication initiation factor DnaB of *B. subtilis* (Rokop *et al*, 2004). Consistent with a role in DNA replication initiation (Li & Araki, 2013), MPN525 (DnaB-like) is required for growth, but is generally not detected in standard growth conditions. This implies that DnaB-like is expressed at a certain moment of the cell cycle. Based on these observations, we hypothesize that Lon-mediated regulation of MPN525 (DnaB-like) may represent a cell cycle checkpoint to restrict and control the frequency and timing of cell division in this slow growth organism.

Despite its reduced genome, *M. pneumoniae* conserves a significant number of genes associated with the restriction-modification (R-M) system. With the exception of the HsdS subunit MPN638, which is highly expressed in normal growth conditions (Maier *et al*, 2011), our study revealed that all HsdS subunits and putative methyltransferases are unstable in the presence of Lon. As MPN638 is the less conserved HsdS subunit (Appendix Fig S9), it is possible that this particular HsdS-like protein may have additional functions. We did not find changes in the levels of the putative restriction enzymes, suggesting that Lon-dependent regulation is mainly associated with the potential ability of this organism to modulate its genome methylation pattern. Although type I R-M systems are commonly referred to as bacterial defense mechanisms against foreign DNA, increasing evidence supports additional roles, including epigenetic changes that alter bacterial phenotypes (Vasu & Nagaraja, 2013; Croix *et al*, 2017). Hence, we envision a model in

which *M. pneumoniae* could potentially change its genome methylation pattern by alternating the formation of diverse HsdS–HsdM complexes in a Lon-dependent manner.

We also addressed whether different substrates share similar Lon or FtsH degradation signals. Sequence analysis of the candidate substrates failed to reveal common features, consistent with previous reports showing that recognition mechanisms of known Lon and FtsH substrates are highly diverse (Tsilibaris *et al*, 2006; Bittner *et al*, 2017). We found that hydrophobic sequences as short as a single residue can promote Lon-dependent degradation of functional proteins in *M. pneumoniae*. In this line, we have recently shown that protein abundances can be influenced by the identity of the last C-terminal amino acid, with hydrophobic residues associated with faster degradation rates (Weber *et al*, 2020). *In vitro* studies performed with the *E. coli* Lon protease has suggested that sequences rich in aromatic residues that are accessible in misfolded proteins act as Lon recognition signals (Gur & Sauer, 2008). Thus, the identified regions in our study likely represent Lon recognition degrons, although we cannot discard that these regions may function as binding sites of putative Lon cofactors required for degradation. How the stability of functional substrates is conditionally regulated is unclear. Possible mechanisms may include protein–protein interactions or conformational changes in the substrate preventing accessibility to degradation signals. Consistent with this, the C-terminal region of FtsZ has been implicated in interactions between FtsZ monomers and FtsA (Ma & Margolin, 1999; Yan *et al*, 2000). Similarly, the amphiphatic helix at the C-terminus of FtsA in *E. coli* is required to localize at the membrane (Pichoff & Lutkenhaus, 2005), suggesting that membrane anchoring could prevent degradation.

Based on our work, we can now present a fully integrated view of how *M. pneumoniae* regulates protein degradation with the minimum number of protease activities required for protein homeostasis (see Fig 7). In fact, almost the entire set of intracellular proteases present in *M. pneumoniae* is essential for cell survival, highlighting its low functional redundancy as compared to other model organisms. Our study also provides novel insights into the cellular response and regulation associated with the main proteases of *M. pneumoniae*. Despite the paucity of protease genes and the apparent lack of protein adaptors, we show that this minimal organism has evolved to coordinate important cellular processes taking advantage of the broad recognition ability of the universal Lon protease.

# Materials and Methods

## Reagents and Tools table

| Reagent/Resource | Reference or Source | Identifier or Catalog Number |
|---|---|---|
| **Experimental models** | | |
| *Mycoplasma pneumoniae* M129 | Richard Herrmann lab | |
| Other mycoplasma strains | This study | Appendix Table S1 |
| *Escherichia coli* TOP10 | Invitrogen | C404003 |
| *Escherichia coli* DH5α | NEB | C2987H |
| **Recombinant DNA** | | |
| Plasmids | This study | Appendix Table S2 |
| **Antibodies** | | |
| Mouse monoclonal anti-BrdU (1:2,000) | Sigma | B2531 |
| Mouse monoclonal anti-FLAG M2 (1:5,000) | Sigma | F1804, F3165 |
| Rabbit polyclonal anti-Firefly luciferase (1:4,000) | Invitrogen | PA5-32209 |
| Rabbit polyclonal anti-CAT (1:2,000) | Abcam | ab50151 |
| Rabbit polyclonal anti-Lon (1:3,000) | Richard Herrmann lab | |
| Rabbit polyclonal anti-FtsH (1:3,000) | Richard Herrmann lab | |
| Rabbit polyclonal anti-HMW1 (1:10,000) | Richard Herrmann lab | |
| Rabbit polyclonal anti-P65 (1:3,000) | Richard Herrmann lab | |
| Rabbit polyclonal anti-P30 (1:10,000) | Richard Herrmann lab | |
| Rabbit polyclonal anti-RL7 (1:5,000) | Richard Herrmann lab | |
| Sheep polyclonal HRP-conjugated anti-mouse IgG (1:10,000) | Sigma | A6782 |
| Goat polyclonal HRP-conjugated anti-rabbit IgG (1:5,000) | Sigma | A0545 |

**Reagents and Tools table** (continued)

| Reagent/Resource | Reference or Source | Identifier or Catalog Number |
|---|---|---|
| **Oligonucleotides and sequence-based reagents** | | |
| Primers | This study | Appendix Table S3 |
| **Chemicals, enzymes and other reagents** | | |
| EcoRV | NEB | R0195S |
| Phusion High-Fidelity DNA Polymerase | Thermo Scientific | F530S |
| X-Gal solution | Thermo Scientific | R0941 |
| Chlorampehnicol | Sigma | C0378 |
| Tetracycline | Sigma | T7660 |
| Gentamycin | Sigma | G1397 |
| HEPES | Sigma | H4034 |
| Sucrose | Sigma | 84097 |
| Urea | Sigma | U5378 |
| Dynabeads™ MyOne Streptavidin C1 | Invitrogen | 65001 |
| Glycogen | Roche | 10901393001 |
| 5-Bromo-2′-deoxyuridine | Sigma | B5002 |
| InstantBlue™ | Expedeon | ISB1L |
| Supersignal West Femto Chemiluminescent Substrate | Thermo Scientific | 34096 |
| Supersignal West Pico Chemiluminescent Substrate | Thermo Scientific | 34080 |
| 13C(6)15N(2)) L-lysine | Sigma | 608041 |
| Digested bovine serum albumin | NEB | P8108S |
| Triton X-100 | Sigma | X100 |
| Thioflavin-T | Sigma | T3516-5G |
| Trypan blue 0.4% solution | Gibco | 15250061 |
| **Software** | | |
| FoldX (v5.0) | Delgado et al (2019) | |
| Proteome Discoverer (v2.0) | Thermo Fisher | |
| Mascot search engine (v2.5) | Matrix Science | |
| Xcalibur software (v2.2) | Thermo Scientific | |
| edgeR (v3.26.8) | Robinson and Smyth (2007, 2008); Robinson et al (2010); Robinson and Oshlack (2010) | |
| SeqPurge tool (v0.1-478-g3c8651b) | Sturm et al (2016) | |
| bowtie2 (v2.3.5) | Langmead and Salzberg (2012) | |
| samtools (v1.9) (using htslib 1.9) | Li et al (2009) | |
| sort (GNU coreutils) 8.26 | | |
| bedtools (v2.27.1) | Quinlan and Hall (2010); Quinlan (2014) | |
| MS-EmpiRe (R package version 0.1.0) | Ammar et al (2019) | |
| eggnog-mapper (v4.5) | Huerta-Cepas et al (2017) | |
| scipy.stats python package (v1.1.0) | Virtanen et al (2020) | |
| SignalP 5.0 web server | Almagro Armenteros et al (2019) | |
| TMHMM server 2.0 | Sonnhammer et al (1998) | |
| seaborn python package (v0.10.1) | | |
| matplotlib python package (v3.2.2) | https://doi.org/10.1109/MCSE.2007.55 | |
| NumPy python package (v1.17.0) | https://doi.org/10.1109/MCSE.2011.37 | |
| pandas python package (v1.0.5) | https://doi.org/10.25080/Majora-92bf1922-00a | |

**Reagents and Tools table**   (continued)

| Reagent/Resource | Reference or Source | Identifier or Catalog Number |
|---|---|---|
| **Other** | | |
| BCA Protein assay Kit | Pierce | 23225 |
| MasterPure DNA purification Kit | Epicentre | MCD85201 |
| Qubit dsDNA HS assay Kit | Invitrogen | Q32854 |
| miRNeasy Mini Kit | Qiagen | 217004 |
| RNase-Free Dnase Set | Qiagen | 79254 |
| Supersignal West Femto Chemiluminescent Substrate | Thermo Scientific | 34096 |
| Supersignal$^{T}$West Pico Chemiluminescent Substrate | Thermo Scientific | 34080 |
| NuPAGE 4–12% Bis-Tris pre-cast polyacrylamide gels | Invitrogen | WG1402BX10 |
| TruSeq Stranded mRNA Sample Prep Kit v2 | Illumina | RS-122-2101 |
| ONE-Glo Luciferase Assay System | Promega | E6110 |
| Amersham Hybond-N+ | GE Healthcare | RPN2222B |
| Qubit Fluorometer | Invitrogen | Q32857 |
| Gene Pulser XCell™ electroporation system | Bio-Rad | |
| Infinite M200 plate reader | Tecan | |
| Tecan Spark plate reader | Tecan | |
| HiSeq 2500 sequencing platform | Illumina | |
| BioAnalyzer | Agilent | |
| UVP CL-1000 Ultraviolet Crosslinker | Analytik Jena | |
| iBlot™ dry blotting system | Invitrogen | |
| LAS-3000 Imaging System | Fujifilm | |
| Bioruptor sonication system | Diagenode | |
| LTQ-Orbitrap Velos Pro mass spectrometer | Thermo Fisher | |

## Methods and Protocols

### Bacterial strains and growth conditions

Wild-type *M. pneumoniae* strain M129 and its derivatives (Reagents and Tools Table, Appendix Table S1) were grown in modified Hayflick medium (Yus *et al*, 2009) at 37°C under 5% $CO_2$ in tissue culture flasks, unless otherwise indicated. When needed, Hayflick medium was supplemented with 0.8% agar, puromycin (3 µg/ml), chloramphenicol (20 µg/ml), or tetracycline (2 µg/ml) for selection of transformants. *Escherichia coli* strain TOP10 (Invitrogen) was used for vector cloning. This strain was grown at 37°C in LB broth or LB agar plates containing ampicillin (100 µg/ml) and X-Gal (40 µg/ml) as needed.

### Construction of Lon and FtsH conditional mutants
#### Construction and mutant design

Lon (ΔIndLon) and FtsH (ΔIndFtsH) conditional mutants were constructed using genome-editing tools mediated by the phage recombinase GP35 as previously described with few modifications (Sun *et al*, 2015; Piñero-Lambea *et al*, 2020). The specific genome editings were performed as follows (see also Appendix Fig S1 for illustration).

In *M. pneumoniae*, Lon (MPN332) transcription is controlled by the trigger factor (MPN331) promoter and a regulatory CIRCE element located upstream of the Lon coding region. To achieve conditional Lon expression, we replaced the CIRCE sequence in the genome by a Tet-inducible platform. This platform was designed to contain a *cat* selectable marker and a *tetR* repressor, both oriented in opposite direction to that of *lon* expression. In addition, lox sites flanked the *cat* selectable marker, allowing *cat* excision by the Cre recombinase (Mariscal *et al*, 2016). The platform also contained the Pxyl/TetO2-inducible promoter (Mariscal *et al*, 2016), which was engineered to be located upstream of the *lon* start codon after genome edition. Importantly, the Pxyl/TetO2 promoter was transcriptionally isolated by placing an endogenous terminator (MPN625) upstream the inducible platform. In order to guide the genome replacement, we enclosed the whole inducible platform by flanking regions of the CIRCE element. As described below, both flanking regions and the inducible platform were produced as a single ssDNA recombineering substrate and transformed into M129_GP35 strain, a wild-type strain expressing the *gp35* gene (Piñero-Lambea *et al*, 2020).

A similar strategy to insert an inducible platform to control the expression of FtsH within the endogenous locus was unsuccessful. This failure may be explained by a low genome-editing efficiency and because *ftsH* (*mpn671*) forms part of a complex transcriptional unit containing six genes, most of which have essential functions and overlapping regions. As an alternative, we used a two-step approach. First, we introduced by transposon delivery the *ftsH* gene under the control of the inducible platform in the M129_GP35

strain, thus generating the M129 + pMTnTc_IndFtsH strain. The transposon vector (pMTnTc_ftsH_Ind) used to generate this strain was obtained by cloning into a pMTnTetM438 vector (Pich *et al*, 2006), the *ftsH*-inducible platform containing the *tetR* and the *ftsH* gene under the control of Pxyl/TetO2 promoter. Molecular cloning was performed by Gibson assembly as detailed in Appendix Table S2 and using the primers listed in Reagents and Tools Table (Appendix Table S3). Then, we deleted the endogenous *ftsH* gene in M129 + pMTnTc_IndFtsH strain by transforming an ssDNA recombineering substrate containing the *cat* selectable marker enclosed by *ftsH* flanking regions. The *cat* selectable marker was flanked by lox sites, allowing *cat* excision by the Cre recombinase. In total, 92% of the endogenous *ftsH* coding sequence was deleted, leaving 50 and 100 bp at the 5′ and 3′ ends, respectively, to preserve overlapping regions with flanking genes. To prevent transcriptional polar effects on the downstream genes, we also included a promoter after the *cat* gene that replaced the *ftsH* endogenous locus. Finally, to determine the simultaneous effect of the absence of both proteases, we also constructed a Lon/FtsH double mutant (ΔIndLon_FtsH). For this, we generated the same genome edition within the *lon* locus in the FtsH-inducible mutant, in which we excised the *cat* selectable marker using a Cre-lox system as previously described (Mariscal *et al*, 2016).

### Production of ssDNA recombineering substrates

The recombineering substrates to perform the genome modifications described above were obtained as follows. For the genome edition of the *lon* locus, both flanking regions and the inducible platform were generated and cloned into a pBSKII+ (Invitrogen) by Gibson assembly generating plasmid pΔLonPr_Ind (Reagents and Tools Table, Appendix Table S2). To delete the endogenous *ftsH* gene, we cloned the *cat* selectable marker enclosed by *ftsH* flanking regions into a pBSKII + by Gibson cloning generating plasmid pΔftsH (Reagents and Tools Table, Appendix Table S2). Primers used during Gibson assembly are listed in Reagents and Tools Table (Appendix Table S3). PCR templates to generate ssDNA recombineering substrates were obtained using pΔLonPr_Ind and pΔftsH plasmids as templates and the pair of primers Bio_lonPr_F/ Pro_lonPr_R or Pro_KOftsH_F/Bio_KOftsH_R, respectively (Reagents and Tools Table, Appendix Table S3). These primers were designed to contain biotin or phosphorothioate modifications attached to the 5′ ends in order to allow ssDNA purification and protection of the ssDNA substrate. To generate ssDNA substrates, 120 μl of Streptavidin dynabeads (MyOne™ Streptavidin C1, Invitrogen) were washed three times with washing buffer (10 mM Tris, 1 mM EDTA, 2 M NaCl, pH 7.5), and incubated with 20 μg of the corresponding PCR product by rotation at RT for 2 h. Dynabeads were then recovered and resuspended in 50 μl melting buffer (125 mM NaOH). After a gentle vortex mixing, magnetic beads were pulled down and the supernatant solution recovered and diluted in 500 μl of neutralization buffer (60 mM NaAc in TE buffer). A second round of elution was performed and recovered to the same neutralization solution. The ssDNA was precipitated by adding 60 μg of glycogen and 1 volume of isopropanol. After 30 min of incubation at RT, ssDNA was recovered by centrifugation (18,000 *g*, 45 min at 4°C), and the pellet washed twice with chilled 70% ethanol. Finally, the pellet was air dried and resuspended in electroporation buffer (8 mM HEPES, 272 mM sucrose, pH 7.4).

### Transformation and isolation of mutants

To obtain Lon and FtsH conditional mutants, M129_GP35 and M129 + pMTnTc_IndFtsH strains were transformed respectively with 3 μg of the corresponding ssDNA recombineering substrate (see above). Bacteria transformation was accomplished by electroporation as previously described (Weber *et al*, 2020). To allow GP35 mediated recombination, electroporated cells were cultured in 25-cm² flasks containing 5 ml of Hayflick medium during 24 h. Then, cells were recovered and mutants selected in Hayflick agar plates containing 20 μg/ml chloramphenicol and 100 ng/ml tetracycline to induce Lon and/or FtsH expression. The intended genetic modifications were confirmed by PCR screening as shown in Appendix Fig S1 using primers listed in Reagents and Tools Table (Appendix Table S2). Genetic editions were further confirmed by RNA-seq mapping. The specific transposon insertion sites in each of the strains were also determined by RNA-seq mapping. In particular, the transposon expressing the *gp35* gene was located in coordinate 613384 in the ΔIndFtsH mutant. In the case of the ΔIndLon mutant, we identified two possible insertion sites, in coordinate 168443 or 493019. Unfortunately, as both regions contain repetitive sequences, we could not discern between both possibilities. Finally, the pMTnTc_ftsH_Ind mintransposon in the ΔIndFtsH mutant was located in the coordinate 372403. The same transposon insertions found in ΔIndFtsH were detected in ΔIndLon_FtsH strain, consistent with the fact that it is a derivative strain.

### Culture conditions for Lon and FtsH depletion

*Mycoplasma pneumoniae* strains ΔIndLon, ΔIndFtsH, and ΔIndLon_FtsH were grown in 5 ml cultures supplemented with 20 μg/ml of chloramphenicol and 10 ng/ml of tetracycline to induce Lon and/or FtsH expression. After 48 h of culture, cells were washed with Hayflick twice and scraped off from the flasks in 5 ml of fresh medium without tetracycline. To grow cells under inducing or depleting conditions, new 5 ml cultures containing plain medium (depletion) or supplemented with 100 ng/ml tetracycline (induction) were inoculated with 1:5 or 1:12.5 dilutions of cell suspension, respectively. Cell samples were processed and analyzed after 48 h (to deplete Lon) or 72 h (to deplete FtsH) of depletion, unless otherwise indicated.

### Growth curve analyses
#### pH growth curve analysis

Preparation of starting inocula and growth curve analyses (based on the "growth index" method) was performed as previously described (Yus *et al*, 2019). Briefly, the conditional mutants were grown in inducing conditions to exponential phase and washed twice in Hayflick medium before inoculation in the presence or absence of tetracycline (100 ng/ml). Growth was then recorded in a Tecan Spark plate reader by determining the growth index value, which is the ratio of absorbance at 430 and 560 nm of the culture medium.

#### pH growth curve analysis after heat stress conditions

ΔIndLon and ΔIndFtsH strains were grown in inducing and depleting conditions as described above during 60 h. Cells from each culture were then resuspended in 5 ml of fresh medium and split in five 1 ml aliquots. One aliquot was kept at 37°C, whereas the others were treated at 45°C during 5, 10, 15, and 20 min, respectively. After treatment, wells of a 96-well plate containing 200 μl of

medium supplemented with 100 ng/ml tetracycline (induction) were inoculated per duplicate with 10 μl of cell suspension. Growth was then measured using the "growth index" method in a Tecan Spark plate reader as described above.

### Protein and DNA cell biomass growth curve analysis

ΔIndLon and ΔIndFtsH strains were cultured in tissue culture flasks of 25 cm² containing 5 ml of medium supplemented with 10 ng/ml of tetracycline. After 48 h of culture, cells were washed with Hayflick twice and scraped off from the flasks in 1 ml of fresh medium without tetracycline. For each strain and time point, new 5 ml cultures containing plain medium (depletion) or supplemented with 100 ng/ml tetracycline (induction) were inoculated with 1:100 dilutions of cell suspension. Cultures were grown and processed in duplicate. Protein and DNA biomass were measured at different time points (0, 24, 48, 72, 96, 120 h) as follows. For each time point, cells were washed with 1 × PBS once and scraped off from the flasks in 1 ml of 1 × PBS. Then, the cell suspension was split in 0.5 ml aliquots to obtain samples for both protein and DNA measurements and harvested by centrifugation (13,100 g, 10 min). For protein biomass quantification, the cell pellet was resuspended in lysis buffer containing 1% SDS and disrupted by sonication using a Bioruptor sonication system (Diagenode) with an On/Off interval time of 30/30 s at high frequency for 10 min. Finally, duplicate protein measurements were performed using the Pierce™ BCA Protein Assay Kit. For DNA biomass quantification, the cell pellet was lysed and the DNA extracted using the MasterPure DNA purification Kit (Epicentre) following the recommendations of the Kit manufacturer. Finally, extracted DNA for each time point was measured using a fluorometric method (Qubit dsDNA HS assay Kit, Invitrogen).

### Dot blot analysis of cellular DNA replication

ΔIndLon and ΔIndFtsH strains were grown in tissue culture flasks of 25 cm² containing 5 ml of medium under inducing or depleting conditions. At different interval time points across the growth curve (0, 24, 48 and 72 h post-inoculation), 100 μM of 5-Bromo-2′-deoxyuridine (BrdU, Sigma) was added to each culture and incubated during 24 h. BrdU pulse-labeled cells were collected and total DNA extracted using the MasterPure DNA purification Kit (Epicentre). Extracted DNA for each time point was quantified using a fluorometric method (Qubit dsDNA HS assay Kit, Invitrogen) and denatured by incubation with 0.5 M NaOH for 20 min at 42°C. A total of 100 ng of DNA was dot blotted on a hybond-N + membrane (GE Healthcare) and fixed by ultraviolet cross-linking at 0.120 J/cm² in a UVP CL-1000 Ultraviolet Crosslinker (Analytik Jena). The membrane was blocked with 5% skim milk (Sigma) in PBS containing 0.1% Tween 20 solution and probed with monoclonal anti-BrdU (Sigma) antibody (1:2,000). Anti-mouse IgG (1:10,000) conjugated to horseradish peroxidase (Sigma) was used as a secondary antibody. Blots were developed with the Supersignal™ West Femto Chemiluminescent Substrate detection Kit (Thermo Scientific) and signals detected in a LAS-3000 Imaging System (Fujifilm).

### RNA sample preparation and RNA-seq analyses

*Mycoplasma pneumoniae* conditional mutants were grown in inducing or depleting conditions as described above per duplicate. Before

RNA isolation, the culture medium was changed with fresh one and the cells further incubated for 6 h. At this point, cells were washed with 1 × PBS and lysed immediately with 700 μl Qiazol (Qiagen). RNA isolation was performed using the miRNeasy kit (Qiagen) following the manufacturer's instructions, including the in-column DNase I treatment. The quality of RNA (amount and integrity) was assessed using a BioAnalyzer (Agilent). RNA-seq libraries were prepared at the CRG ultrasequencing facility using the TruSeq Stranded mRNA Sample Prep Kit v2 according to the manufacturer's protocol using the following modifications. The poly(A) selection step was omitted, and fragmentation was done using 100 ng total RNA as starting material. To maintain smaller library insert sizes than in the standard protocol, the first AMPure XP purification after adaptor ligation was performed using 50 μl AMPure XP beads instead of 42 μl. The second round of bead purification was then performed using 55 μl AMPure XP beads instead of 50 μl. The purification of the PCR after library amplification was done using 55 μl AMPure XP beads instead of 50 μl. Sequencing was performed using a HiSeq 2500 (Illumina) with HiSeq v4 chemistry and 2 × 50 bp paired-end reads.

Processing of sequencing reads was performed as follows. Adapter sequences were trimmed from short paired-end reads by using the SeqPurge tool (version 0.1-478-g3c8651b) (Sturm et al, 2016), keeping trimmed reads with a minimum length of 12. Reads were aligned to the wild-type genome of *M. pneumoniae M129* (NCBI accession NC_000912.1) and to the transposon insert sequences using bowtie2 v. 2.3.5 (Langmead & Salzberg, 2012), with parameters values: end-to-end mode, 0 mismatches (-N), seed length of 20 nt (-L), very sensitive mode (-L 20 -D 20 -R 3 -i 'S, 1, 0.50'), maximum fragment length 1,200 nt (-X), only best alignment reported (-k 0). Alignment files were converted from SAM format to sorted indexed BAM format using samtools v. 1.9 (using htslib 1.9) (Li et al, 2009) and sort (GNU coreutils) 8.26. Reads were further filtered by a minimum quality (MAPQ) threshold of 15, keeping only primary and mapped reads, and converted to sorted BEDPE format using samtools and bedtools v2.27.1 (Quinlan & Hall, 2010; Quinlan, 2014). Fragment counts per annotation region were computed using bedtools, with strand-specific overlaps with minimum overlap fraction of 0.5 of read length. Finally, strand-specific per-base coverage was computed using bedtools.

Differential expression analysis was performed using edgeR v. 3.26.8 (Robinson & Smyth, 2007, 2008; Robinson et al, 2010; Robinson & Oshlack, 2010), using trimmed mean of *M*-values (TMM) normalization, and classical pair-wise comparison between induced and depleted conditions. Significance of the fold changes was tested by the exact test based on dispersions estimated by the quantile-adjusted conditional maximum-likelihood (qCML) method. Multiple tests were corrected using Benjamini–Hochberg method (Benjamini & Hochberg, 1995), and fold changes with a corrected *q*-value smaller than 0.05 were considered significant (5% FDR).

### Protein sample preparation of mutant strains for Mass spectrometry analysis

*Mycoplasma pneumoniae* conditional mutants were grown in inducing or depletion conditions as described above per duplicate. Before cell lysis, the culture medium was changed with fresh one and the cells further incubated for 6 h. At this point, cells were washed with 1 × PBS twice, scraped off from the flasks, and

centrifuged at 13,100 $g$ for 10 min. The pellet was resuspended in lysis buffer (4% SDS, 100 mM HEPES, pH 7.4) and the cell lysate disrupted using a Bioruptor sonication system (Diagenode) with an On/Off interval time of 30/30 s at high frequency for 10 min. Finally, cell lysates were spun, and the extracted protein quantified using the Pierce BCA Protein Assay Kit before mass spectrometry analysis.

### Proteome-wide measurements of protein half-lives
#### Pulse-chase experiments and SILAC labeling
Protein turnover rates of *M. pneumoniae* proteins were measured by pulse-chase experiments following the analysis method for SILAC as described in Pratt *et al* (2002) (see also Schwanhäusser *et al*, 2011; Christiano *et al*, 2014). Briefly, *M. pneumoniae* was passaged several times in Hayflick medium supplemented with 20 mM heavy (Lys-8, 13C(6)15N(2)) L-lysine until reaching a labeling efficiency of 86%. Then, labeled cells were washed three times with Hayflick before being transferred in duplicate in unlabeled medium. After 0.5, 3, 6, 8, 12, 18, 24, 30, and 36 h of culture, cells were washed three times in 1 × PBS and harvested by centrifugation. For the longer time points, medium was changed every 12 h. Samples before cell inoculation into unlabeled medium (0 h) were also taken as a reference. Finally, the pellets were resuspended in lysis buffer (7 M urea, 0.2 M NH$_4$HCO$_3$ or 4% SDS, 100 mM HEPES, pH 7.4) and protein concentration determined using the Pierce™ BCA Protein Assay Kit before mass spectrometry analysis. Protein half-lives were estimated from two to four biological replicates.

To take into account the dilution rate due to cell growth, we also computed the growth rate of the cells in exponential phase by measuring protein amounts along the growth using the Pierce™ BCA Protein Assay Kit. Exponential fit of time points at 3, 6, 8, 12, 18, 24, 30, 36, and 48 h measured by duplicate yielded an average doubling time of 8.63 ± 0.13 h (standard deviation).

#### Data analysis
Data analysis of SILAC experiments was performed as follows. Peptide quantification data were retrieved from the Precursor ions quantifier node from Proteome Discoverer (v2.0), and protein ratios were calculated as the mean of peptide group ratios. To estimate the degradation rate, we measure the increase in light protein compared with heavy protein labeling along the growth curve. From time zero, we assumed that heavy protein production is zero and that production of light protein is constant (constitutive):

$$\frac{dH}{dt} = -dH(t)$$

$$\frac{dL}{dt} = \beta - dL(t)$$

where $d$ is the apparent degradation rate, the sum of the protein degradation rate $d_P$ and the dilution rate $\alpha$ due to cell growth. The solution to these ODEs reads,

$$H(t) = H^0 e^{-dt}$$
$$L(t) = \frac{1}{d} e^{-dt}(dL^0 - \beta + \beta e^{dt})$$

We assumed that the cells are at steady state both before and during the whole course of the experiment, such that the total (light + heavy) protein amount, $A(t) = L(t) + H(t)$, remains constant with time,

$$\frac{dA}{dt} = \beta - dA(t) = 0$$

$$A^{s.s.} = \frac{\beta}{d} = L^0 + H^0$$

We defined the relative isotope abundance (RIA) as,

$$RIA(t) = \frac{H(t)}{L(t) + H(t)}$$

using the steady-state assumption, $\beta/d = L^0 + H^0$, and the solution for $H(t)$ and $L(t)$,

$$RIA(t) = \frac{H^0}{L^0 + H^0} e^{-dt}$$

The ratio of the light to heavy protein concentration can similarly be expressed as,

$$r_{LH}(t) = \frac{L^0 + H^0}{H^0} e^{dt} - 1$$

Thus, we can use linear regression on $ln(r_{LH} + 1)$ to estimate both the apparent degradation rate and the estimated initial labeling,

$$ln(r_{LH} + 1) = ln(\frac{L^0 + H^0}{H^0}) + dt$$

For each protein time series of the light to heavy ratios, we performed linear regression using the curve_fit least squares method of the python scipy.optimize package v1.1.0 with bounds on parameters $d \in [0,5]$, $RIA^0 \in [0.5,1]$. Standard errors on the fitted parameters, $\sigma_d$ and $\sigma_{RIA^0}$, were estimated as the square root of the diagonal elements of the variance–covariance matrix.

We observed that for some proteins, the ratio at 36 h was lower than expected. This could be due to the cells entering early stationary phase, or to the larger variability in the heavy protein quantification by mass spectrometry, whose level decreases rapidly close to the detection limit. We performed the regression for the time series with and without the 36-h time point, and selected the fit with the lowest standard deviation in parameter $d$. In addition, we required at least three time points for a valid regression. In total, a successful fit could be obtained for 488 proteins. The apparent degradation rate $d$ is the sum of the protein degradation rate $d_P$ and the dilution rate due to cell growth $\alpha$. We can therefore express the protein degradation rate as $d_P = d - \alpha$. Variability in the apparent degradation rate and growth rate was propagated to the protein degradation rate assuming normally distributed measurements, such that $\sigma_{d_p}^2 = \sigma_\alpha^2 + \sigma_d^2$. We then assumed that the protein degradation rate of each replicate $i$ was drawn from independent normal distributions with the same mean $\mu$ but different variances $\sigma_i^2$. In order to compute the average protein degradation rate over the four replicates, we used the weighted mean

$$\overline{d_P} = \sum_{i=1}^{4} \omega_i d_{P,i}$$

with weights equal to the inverse of the variance of each replicate,

$$\omega_i = \frac{\sum_j \sigma_j^2}{\sigma_i^2}$$

where we normalized the weights to unity. These weights are called reliability weights and intuitively give more weight to data point with smaller variance. We then used the following estimator for the corrected variance of the weighted mean,

$$s^2 = \frac{\sum_i \omega_i}{\left(\sum_i \omega_i\right)^2 - \sum_i \omega_i^2} \sum_i \omega_i (d_{P,i} - \overline{d_P})^2$$

Many proteins exhibited a fitted apparent degradation rate $d$ very close to the dilution rate $\alpha$, such that the protein degradation rate $d_P$ was very small (long protein half-life). For those cases, the apparent degradation rate can be approximately equal or even smaller than the dilution rate due to the experimental variability, resulting in a negative computed protein degradation rate (43 proteins). In this case, we cannot deduce a precise value for the protein degradation rate $d_P = d - \alpha$. However, we can assume that the protein degradation rate is close to zero given its standard error. In order to simplify downstream analyses, we fixed the half-lives of those proteins to a maximum value of 300 h ($d_P = 2.31 \times 10^{-3}$), which reflected the smallest degradation rate measurable given the experimental variability.

### Mass spectrometry analyses

Samples lysed with urea or SDS were digested with trypsin in solution or using the filter-aided protocol (FASP) (Wiśniewski et al, 2009), respectively. Samples were analyzed using a LTQ-Orbitrap Velos Pro mass spectrometer (Thermo Fisher Scientific, San Jose, CA, USA) coupled to an EASY-nLC 1000 (Thermo Fisher Scientific (Proxeon), Odense, Denmark). Peptides (1 µg for mutant strains and 2 µg for SILAC samples) were loaded onto the 2-cm Nano Trap column with an inner diameter of 100 µm packed with C18 particles of 5 µm particle size (Thermo Fisher Scientific) and were separated by reversed-phase chromatography using a 25-cm column with an inner diameter of 75 µm, packed with 1.9 µm C18 particles (Nikkyo Technos Co., Ltd. Japan). Chromatographic gradients started at 93% buffer A and 7% buffer B with a flow rate of 250 nl/min for 5 min and gradually increased 65% buffer A and 35% buffer B in 120 min. After each analysis, the column was washed for 15 min with 10% buffer A and 90% buffer B. Buffer A: 0.1% formic acid in water. Buffer B: 0.1% formic acid in acetonitrile.

The mass spectrometer was operated in positive ionization mode with nanospray voltage set at 2.1 kV and source temperature at 300°C. Ultramark 1621 was used for external calibration of the FT mass analyzer prior analyses, and an internal calibration was performed using the background polysiloxane ion signal at $m/z$ 445.1200. The acquisition was performed in data-dependent acquisition (DDA) mode, and full MS scans with one microscans at resolution of 60,000 were used over a mass range of $m/z$ 350–2,000 with detection in the Orbitrap. Auto gain control (AGC) was set to 1E6, dynamic exclusion (60 s) and charge state filtering disqualifying

singly charged peptides was activated. In each cycle of DDA analysis, following each survey scan, the top twenty most intense ions with multiple charged ions above a threshold ion count of 5,000 were selected for fragmentation. Fragment ion spectra were produced via collision-induced dissociation (CID) at normalized collision energy of 35% and they were acquired in the ion trap mass analyzer. AGC was set to 1E4, and isolation window of 2.0 $m/z$, an activation time of 10 ms, and a maximum injection time of 100 ms were used. All data were acquired with Xcalibur software v2.2. Digested bovine serum albumin (New England Biolabs) was analyzed between each sample to avoid sample carryover and to assure stability of the instrument and QCloud (Chiva et al, 2018) has been used to control instrument longitudinal performance during the project.

Acquired spectra were analyzed using the Proteome Discoverer software suite (v2.0, Thermo Fisher Scientific) and the Mascot search engine v2.5 Matrix Science (Perkins et al, 1999). The data were searched against a home-made database consisting on a list of common contaminants and all possible M. pneumoniae ORFs > 19 aa (87,051 entries). For peptide identification, a precursor ion mass tolerance of 7 ppm was used for MS1 level, trypsin was chosen as enzyme and up to three missed cleavages were allowed. The fragment ion mass tolerance was set to 0.5 Da for MS2 spectra. Oxidation of methionine and N-terminal protein acetylation were used as variable modifications whereas carbamidomethylation on cysteines was set as a fixed modification. For SILAC samples, Lys-8 (13C(6) 15N(2)) was also used as variable modification. False discovery rate (FDR) in peptide identification was set to a maximum of 5%. Peptide quantification data were retrieved from the "Precursor ion area detector" node from Proteome Discoverer (v2.0) using 2 ppm mass tolerance for the peptide extracted ion current (XIC).

### Statistical analyses and criteria to define substrate candidates

Statistical analysis of the differences in protein abundances between samples and identification of substrate candidates was performed in two stages. In the following, we describe peptides with the notation ($n^+$-$n^-$), where n is the number of replicates in which the peptide was detected and its signal quantified, and +/− indicates the condition analyzed (+ for induced, − for depleted).

In the first stage, we identified differentially expressed (DE) proteins. We estimated protein fold changes for those proteins which had at least one common unique peptide detected in all the replicates of the two conditions (2-2 peptide), using the peptide-based statistical method MS-EmpiRe (Ammar et al, 2019) (R package version 0.1.0). Peptide-based regression methods for analyzing label-free quantitative proteomics data were shown to outperform summarization-based pipelines (such as the TOP3 method) on benchmark datasets (Goeminne et al, 2015). In the MS-EmpiRe pipeline, the significance of peptide fold changes is tested against a signal strength-dependent empirical error distribution computed from the collection of all peptide fold changes across replicate pairs within the same condition. In order to obtain a comparison of protein levels between two conditions, peptide fold changes for the same protein and their significance scores are combined, taking into account the empirical error distribution of each peptide. Finally, Benjamini–Hochberg method (Benjamini & Hochberg, 1995) is applied to correct for multiple tests. Normalization and outlier detection methods were used as described in the

pipeline with default parameters. Following this method, protein changes were computed in the three conditional mutants by comparing induced to depleted conditions, with two replicates in each condition. Proteins with an adjusted *P*-value below 5% FDR and with a $\log_2$(fold change) larger than 1 (smaller than −1) were considered as upregulated (downregulated). In order to distinguish changes due to transcriptional regulation from changes due to differential degradation rate, we integrated changes in mRNA levels by computing the ratio of protein fold change to mRNA fold change. Differentially expressed (DE) proteins were defined as the proteins which were upregulated at the protein level and exhibited protein/mRNA fold change ratio higher than 2-fold ($\log_2$(protein_FC/ mRNA_FC) ≥ 1).

In the second stage, we identified differentially detected (DD) proteins. The aforementioned DE analysis only considered proteins with at least one shared peptide detected in all samples (2-2 peptide), thereby filtering out many proteins of low abundance. In this analysis stage, we relaxed these criteria in order to recover potential substrates. We reasoned that substrate proteins which were not detected in the induced condition due to their low abundance would substantially increase in the depleted condition and get detected. In order to discern true protein abundance increase from randomness in peptide detection, we selected proteins which had at least one shared peptide not detected in any of the two induced samples and detected in both depleted samples (0-2 peptide). In order to take into account transcriptional changes, we further selected proteins whose corresponding mRNA fold change was lower than 2 ($\log_2$(mRNA_FC) ≤ 1).

### Analysis of protein aggregates

Protein aggregation was analyzed after Triton X-100 solubilization as previously described with some modifications (Maisonneuve *et al*, 2008). Briefly, ΔIndLon and ΔIndFtsH strains were grown in inducing and depleting conditions as described above in 20 ml cultures. After 48 h (Lon) or 72 h (FtsH) of depletion, cells were washed with 1 × PBS thrice and lysed directly into the flask with a lysis buffer containing 1% Triton X-100 in 1 × PBS. Lysed cells were collected and incubated at 4°C on a rotating platform for 1 h. Total cell lysates were normalized by protein quantification using the Pierce™ BCA Protein Assay Kit, prior centrifugation at 20,000 *g* for 15 min. Insoluble fractions were then processed for Thioflavin-T (ThT) staining or MS analysis as follows.

For ThT binding assays, insoluble fractions were resuspended in 1 × PBS containing 100 µM ThT and fluorescence recorded at an emission wavelength of 581 nm using 440 nm excitation in a Tecan Infinite M200 plate reader. Background fluorescence was subtracted from all readings. Data were obtained from three biological replicates. In parallel experiments, the insoluble fractions were resuspended in lysis buffer (4% SDS, 100 mM HEPES, pH 7.4) and homogenized by sonication using a Bioruptor sonication system (Diagenode) with an On/Off interval time of 30/30 s at high frequency for 10 min. Proteins were quantified using the Pierce™ BCA Protein Assay Kit, and two biological replicates were subjected for LC/MS analysis as described above. Statistical analysis of proteins enriched in the insoluble fraction was performed using the same peptide-based approach as described above. Protein changes were computed by comparing the whole cell lysate samples obtained in the previous experiment to the insoluble fractions

samples. The same analysis was performed separately for the induced and depleted conditions, with two replicates for each condition.

### Cell membrane integrity assay

Membrane damage was assessed by a trypan blue dye-exclusion assay as previously described with some modifications (Uliasz & Hewett, 2000). ΔIndLon and ΔIndFtsH strains were grown in inducing and depleting conditions in 24-well plates containing 1 ml of medium per well. After 72 h of depletion, induced and depleted cells were washed with 1 × PBS twice and triplicate wells for each condition and strain were untreated or exposed during 30 min at room temperature to 1 × PBS containing 0.001% Triton X-100 (mild solubilization treatment), or 0.01% Triton X-100 (harsh solubilization treatment). Cells were then stained for 2 min with a 0.2% solution of Trypan blue (Gibco), and gently washed with 1 × PBS thrice. After removing the excess of dye, cells were directly lysed with 150 µl of 1% SDS, and 100 µl of the SDS/trypan blue solution gently transferred to a 96-well plate avoiding the addition of bubbles. Finally, absorbance at 590 nm was recorded in a Tecan Infinite M200 plate reader. Non-stained wells were used to determine background values, which were subtracted from all readings. To account for differences in the amount of cells present in each well, measurements were normalized by the signal obtained with the 0.01% Triton X-100 treatment. These conditions fully permeabilize the cell membrane and give a linear relationship between signal and cell amount. For each strain and condition, data were collected from three independent wells per experiment (considered technical replicates) of a total of three separate biological replicate experiments.

### Validation of candidate substrates

To confirm Lon and FtsH substrates, we generated N- and C-terminal FLAG variants of selected candidates. Similarly, we also constructed FLAG variants with mutations in selected putative degrons. All gene variants were generated by PCR amplification as detailed in Appendix Table S2 using the primers listed in Reagents and Tools Table (Appendix Table S3). Lon substrate FLAG variants were cloned by Gibson assembly into pMTnTetM438 vector (Pich *et al*, 2006), whereas FLAG variants of FtsH substrates were cloned into pMTnCat vector (Burgos *et al*, 2012). All FLAG variants were expressed from the P438 promoter (Pich *et al*, 2006). The resulting transposon vectors were transformed into the corresponding strain (ΔIndLon or ΔIndFtsH) by electroporation as previously described (Weber *et al*, 2020). To allow selection of transformants in the ΔIndFtsH strain, we excised the *cat* resistance marker by Cre recombination as previously described (Mariscal *et al*, 2016). To assess the stability of the FLAG variants, transformant pools were grown in inducing or depleting conditions (48 and 72 h for Lon and FtsH, respectively) as described above. Then, cell lysates were prepared as follows. Cells were washed with 1 × PBS twice, scraped off from the flasks, and centrifuged at 13,100 *g* for 10 min. The cell pellet was resuspended in lysis buffer containing 1% SDS and disrupted using a Bioruptor sonication system (Diagenode) with an On/Off interval time of 30/30 s at high frequency for 5 min. Finally, substrate stability was assessed by immunoblot using anti-FLAG antibodies.

For some FLAG-tagged variants, we performed additional time course depletion experiments and *in vivo* degradation assays. For the time course depletion experiments, mutants expressing the

FLAG-tagged variants were grown in depleting conditions in a 24-well plate format and cells processed for immunoblot analysis at different time points after removing the inducer. For the *in vivo* degradation assays, mutants expressing the FLAG-tagged variants were grown in depleting conditions in a 24-well plate format for 36 h (Lon substrates) or 60 h (FtsH substrates), and then Lon or FtsH expression transiently induced for 3 h. Afterward, protein synthesis was blocked with gentamicin (100 μg/ml) and cells lysed at different time points and processed for immunoblot analysis as described above. As controls, non-induced wells were also treated with gentamicin and processed at the same time points.

### Construction of an unstable variant of the luciferase reporter

We have recently shown that the firefly luciferase (Fluc) reporter is functional in *M. pneumoniae* (Weber *et al*, 2020). An unstable variant of Fluc protein was designed using FoldX (version 5.0) (Delgado *et al*, 2019). As a template we used the crystal structure (PDB: 1LCI) of Fluc, which was repaired with the command RepairPDB to solve eventual crystallographic errors. To model amino acid substitutions and quantitatively predict their impact on protein stability, we run the command BuildModel that predicted a $\Delta\Delta G$ of 3.228 kcal/mol for the F89E mutation. Disruption of activity of the Fluc variant was confirmed by using the ONE-Glo™ Luciferase Assay System (Promega) as previously described (Weber *et al*, 2020). Primers and Gibson cloning strategy to generate the F89E luc variant are detailed in Reagents and Tools Table (Appendix Tables S3 and S2).

### Construction of HMW2 deletion mutant

A deletion mutant of *hmw2* (*mpn310*) was obtained by GP35 recombination following the procedures described for the conditional mutants. In total, 1,090 bp corresponding to the N-terminal region of the endogenous *hmw2* gene was deleted. To perform this deletion, we constructed the pΔNthmw2 vector by cloning the *cat* selectable marker enclosed by *hmw2* flanking regions into a pBSKII+ by Gibson assembly. Cloning strategy and primers used are shown in Reagents and Tools Table (Appendix Tables S2 and S3). The dsDNA template to generate the ssDNA recombineering substrate was obtained by PCR amplification using plasmid pΔNthmw2 and the pair of primers Bio_KOhmw2_F/Pro_KOhmw2_R (Reagents and Tools Table, Appendix Table S3). Purification of the ssDNA substrate and its transformation in ΔIndLon strain (in which we excised the *cat* selectable marker using a Cre-lox system) was performed as described above for the conditional mutants. Mutants were selected in Hayflick agar plates containing 20 μg/ml chloramphenicol and 100 ng/ml tetracycline to induce Lon expression. The intended genetic modifications were confirmed by PCR screening using primers listed in Reagents and Tools Table (Appendix Table S3).

### SDS–PAGE and immunoblot analyses

Mycoplasma cell lysates were quantified using the Pierce™ BCA Protein Assay Kit, and 10 μg of cell extracts was subjected to electrophoresis through NuPAGE™ 4–12% Bis-Tris pre-cast polyacrylamide gels (Invitrogen). Proteins were then visualized after InstantBlue™ (Expedeon) Commassie staining or transferred onto nitrocellulose membranes using an iBlot™ dry blotting system (Invitrogen). When possible, membranes were cut at appropriate molecular weights and probed with different antibodies independently. For immunodetection, membranes were blocked with 5% skim milk (Sigma) in PBS containing 0.1% Tween 20 solution and probed with monoclonal anti-FLAG M2 (Sigma) antibody (1:5,000), polyclonal anti-Fluc (Invitrogen) antibody (1:4,000), or polyclonal antibodies specific to mycoplasma proteins (kind gift of Dr. Herrmann, Heidelberg University). These include anti-Lon (1:3,000), anti-FtsH (1:3,000), anti-HMW1 (1:10,000), anti-P65 (1:3,000), anti-P30 (1:10,000), and anti-RL-7 (1:5,000). As a loading control, we used a polyclonal anti-CAT (Abcam) antibody (1:2,000). Anti-mouse IgG (1:10,000) or anti-rabbit IgG (1:5,000) conjugated to horseradish peroxidase (Sigma) was used as a secondary antibody. Blots were developed with the Supersignal™ West Pico Chemiluminescent Substrate detection Kit (Thermo Scientific) and signals detected in a LAS-3000 Imaging System (Fujifilm).

### Bioinformatic and statistics analysis

#### Functional enrichment analysis

Genes were classified into functional COG categories using the eggnog-mapper tool (Huerta-Cepas *et al*, 2017) based on eggNOG 4.5 orthology data (Huerta-Cepas *et al*, 2016). Enrichment of COG categories was tested by two-sided Fisher test (scipy.stats python package v1.1.0) (Virtanen *et al*, 2020) followed by multiple test corrections using Benjamini–Hochberg method (Benjamini & Hochberg, 1995) with 5% family-wise false discovery rate (FDR).

#### Enrichment analysis of truncated gene variants

Putative truncated gene variants including pseudogenes and split genes were identified by manual inspection of the genome (Table EV4). To test the enrichment of these gene variants among Lon target candidates, we first discarded genes with very low mRNA levels, as those are unlikely to produce protein at detectable levels. In order to compare mRNA levels, we computed transcripts per million (TPM) from the TMM normalized counts. We selected genes whose average TPM value in the two replicates of the Lon-induced samples was above the $10^{th}$ quantile of all TPM values, resulting in 640 genes. Within this filtered list of genes, truncated genes were found to be enriched in substrate candidates (six truncated genes out of 54) compared with non-substrate genes (21 truncated genes out of 586). Enrichment was tested by means of two-sided Fisher test (scipy package version 1.1.0).

#### Enrichment analysis of membrane proteins

Protein localization was predicted using a combination of prediction tools. First, the presence of signal peptide was detected using SignalP 5.0 web server (Almagro Armenteros *et al*, 2019). Then, presence of transmembrane helices was predicted using TMHMM Server 2.0 (Sonnhammer *et al*, 1998). Proteins possessing either a signal peptide or at least one transmembrane helix were classified as "membrane" proteins (198 out of 737).

#### Correlation analysis of transcriptional changes across experimental perturbations

To examine the transcriptional co-regulation of chaperones and proteases, we performed a correlation analysis of the fold changes observed in the set of 35 environmental and genetic perturbations from Yus *et al* (2019). We excluded those perturbations that induce RNA degradation, such as novobiocin treatment. Clustered heatmaps were drawn using the seaborn python package v0.10.1. Genes

were clustered by similarity in their transcriptional changes or correlation pattern using the Euclidean distance.

### Epistasis analysis

We investigated to which extent the changes in protein and mRNA abundance in the double mutant strain could be explained by the combination of the changes of the two individual mutants. We hypothesized that in the absence of any direct or indirect interaction between the action of the two proteases, the changes in protein to mRNA ratio in the double mutant would be predicted by the sum of the $\log_2$(ratio) in each individual mutant. We thus defined the epistasis as

$$\omega = \log_2(P_{FC}/M_{FC})_{\Delta IndLonFtsH} - (\log_2(P_{FC}/M_{FC})_{\Delta IndLon} + \log_2(P_{FC}/M_{FC})_{\Delta IndFtsH})$$

In order to evaluate which genes presented an epistasis value deviating significantly from zero, we first roughly estimated the noise in $\omega$ as follows. We first estimated the noise in protein abundance measurement by selecting proteins with non-significant protein fold change (based on corrected $P$-value only) and computed its variance, for each of the three experiments, $\sigma^2_{\Delta IndLon}(\log_2 P_{FC})$, $\sigma^2_{\Delta IndFtsH}(\log_2 P_{FC})$ and $\sigma^2_{\Delta IndLonFtsH}(\log_2 P_{FC})$. Similarly for the noise in mRNA, selecting protein with non-significant change in mRNA levels, $\sigma^2_{\Delta IndLon}(\log_2 M_{FC})$, $\sigma^2_{\Delta IndFtsH}(\log_2 M_{FC})$, and $\sigma^2_{\Delta IndLonFtsH}(\log_2 M_{FC})$. We finally estimated the noise in the epistasis observable by assuming a normal distribution of both mRNA and protein measurement noises, such that the variances of all measurements sum up as

$$\sigma^2(\omega) = \sigma^2_{\Delta IndLon}(\log_2 P_{FC}) + \sigma^2_{\Delta IndLon}(\log_2 M_{FC}) + \sigma^2_{\Delta IndFtsH}(\log_2 P_{FC})$$
$$+ \sigma^2_{\Delta IndFtsH}(\log_2 M_{FC}) + \sigma^2_{\Delta IndLonFtsH}(\log_2 P_{FC}) + \sigma^2_{\Delta IndLonFtsH}(\log_2 M_{FC})$$

Proteins for which the absolute value of the epistasis was above a threshold $\omega_{thresh}$ were considered significant, where $\omega_{thresh} = (1/\sigma_{\Delta IndLon}(\log_2 ratio)) * \sigma(\omega)$, such that the Z-score of the epistasis threshold was identical to the Z-score of the ratio threshold of 1 $\log_2$ used in the definition of the substrate candidates.

# Data availability

The RNA sequencing data have been deposited in ArrayExpress repository with the dataset identifier E-MTAB-8537. The mass spectrometry proteomics data have been deposited to the ProteomeXchange Consortium via the PRIDE (Vizcaíno *et al*, 2016) partner repository with the dataset identifiers PXD016343; http://www.ebi.ac.uk/pride/archive/projects/PXD016343 (comparative proteomic study), PXD016386; http://www.ebi.ac.uk/pride/archive/projects/PXD016386 (proteome-wide measurements of protein half-lives), and PXD021506; http://www.ebi.ac.uk/pride/archive/projects/PXD021506 (protein insoluble fraction analysis).

Expanded View for this article is available online.

## Acknowledgements

This work has been supported by the European Research Council (ERC) under the European Union's Horizon 2020 research and innovation program, under grant agreement 670216 (MYCOCHASSIS). We also acknowledge the support of the Spanish Ministry of Science and Innovation to the EMBL partnership, the Centro de Excelencia Severo Ochoa, and the CERCA Program from the Generalitat de Catalunya. M.W acknowledges the European Union's Horizon 2020 research and innovation program under grant agreement 634942 (MycoSynVac) and M.L.-S. The support from FEDER project from Instituto Carlos III (ISCIII, Acción Estratégica en Salud 2016) (reference CP16/00094). We also acknowledge the staff of the CRG Genomics Unit for performing RNA-seq library preparation and sequencing. The proteomics analyses were performed in the CRG/UPF Proteomics Unit which is part of the Spanish Infrastructure for Omics Technologies (ICTS OmicsTech), and it is a member of the ProteoRed PRB3 consortium which is supported by grant PT17/0019 of the PE I + D+i 2013-2016 from the Instituto de Salud Carlos III (ISCIII) and ERDF. We also want to thank Dr. Yus for her guidance in performing SILAC experiments, Dr. Piñero-Lambea for his guidance in the purification of ssDNA substrates and the construction of mutants mediated by GP35 recombination, Damiano Cianferoni for his help in FoldX analysis, and Professor Herrmann for sharing antibodies.

## Author contributions

RB and LS conceived and designed the study. RB design and performed the experiments. SM performed pulse-chase experiments and SILAC labeling. MW conceived and performed all the bioinformatic and statistical analyses. RB, MW, and LS analyzed and interpreted the data. LS and ML-S provided direct supervision. RB, MW, and LS wrote the manuscript and designed the figures. All authors read and approved the final manuscript.

## Conflict of interest

The authors declare that they have no conflict of interest.

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
