## [Review Process File · Molecular Systems Biology]

Protein quality control and regulated proteolysis in the genome-reduced organism *Mycoplasma pneumoniae*

Raul Burgos, Marc Weber, Sira Martinez, Maria Lluch-Senar, and Luis Serrano
DOI: [10.15252/msb.20209530](https://doi.org/10.15252/msb.20209530)

Corresponding author(s): Luis Serrano (luis.serrano@crg.eu)

Review Timeline:

Submission Date:	19th Feb 20
Editorial Decision:	29th Apr 20
Revision Received:	25th Sep 20
Editorial Decision:	29th Oct 20
Revision Received:	4th Nov 20
Accepted:	8th Nov 20

Editor: Jingyi Hou

Transaction Report:

Thank you again for submitting your work to Molecular Systems Biology. We have now heard back from two of the three reviewers who agreed to evaluate your manuscript. Unfortunately, after a series of reminders we did not manage to obtain a report from reviewer #1. In the interest of time, and since the recommendations from the other two reviewers are quite similar, I prefer to make a decision now rather than further delaying the process. As you will see from the reports below, the reviewers acknowledge the potential interest of the study. They raise however a series of concerns, which we would ask you to address in a major revision.

I think that the reviewers' recommendations are rather clear and there is therefore no need for me to reiterate the comments listed below. In particular, both reviewers raise concerns with regards to the "in vivo degradation assay", which needs to be convincingly addressed. Further, additional experiments and analyses are required in order to enhance the conceptual novelty and the level of biological insight provided by the study, as recommended by reviewer #3.

All other issues raised by the reviewers need to be satisfactorily addressed as well. As you may already know, our editorial policy allows in principle a single round of major revision and it is therefore essential to provide responses to the reviewers' comments that are as complete as possible. Please feel free to contact me in case you would like to discuss in further detail any of the issues raised by the reviewers.

Reviewer #2:

The very interesting manuscript by Burgos et al investigate protein degradation in a genome reduced model organism, *Mycoplasma pneumoniae*, which encodes only two ATP dependent AAA + proteases Lon and FtsH. Such AAA+ protease complexes are very important both for regulatory and general proteolysis, which is reflected by the apparent essentiality of lon and ftsH genes in this organism.

To study the effect of these genes on the transcriptome and proteome they constructed strains with lon or ftsH or both under the control of a Tetracycline controlled promoter, allowing the expression of these genes only in the presence of Tetracycline, and therefore also deplete the proteins encoded by these essential genes in the absence of tetracycline (Fig 1).

In the next paragraph on p6 the authors also mention that they measure proteome wide protein half-life times at 13 and 52h (Supposedly shown in the excel sheet which is probably Table S1?; and maybe related to Fig2D?).

Furthermore, to identify the possible protein substrate of the respective protease, they perform quantitative MS (Fig S4) and RNAseq experiments (Fig S11?) in Lon and FtsH depletion strains and correlate the observed transcriptional and protein abundance changes comparing the depletion condition with the complemented strains (Fig 2A). Here they observe upregulated proteins which they classify as differentially detected (DD, proteins not detected in the induced condition and detected in the depleted condition, and whose mRNA level did not increase more than 2-fold) or differentially expressed (DE, significant increase in abundance upon protease depletion that could not be attributed to an increase in mRNA levels). In addition, they also observe proteins which were downregulated upon Lon or FtsH depletion.

Interestingly when analyzing their candidate DD and DE proteins (Table S3) they observe separate and distinct putative substrate proteins, properties and characteristics, also due to the inherent different localization of FtsH as a membrane protein and the more cytosolic Lon (Fig 2 p7) And the cellular response to Lon vs FtsH depletion appears to be distinct (p9/10 Fig S11 or 12)

Then they go on and Flag tag specific candidate protein substrates to confirm again their degradation but the addition of the Flagtag itself is apparently interfering with the in vivo stability (Fig 3), which is well known in the field and also the characterization of possible motifs for the Lon recognition with this approach is not very conclusive (Fig 4)

The following analysis of Lon as a quality control protease is interesting and more convincing (p 10 Fig 5) and the discussion is very interesting and insightful.

-Major comments (C) and questions (Q)

C The usage of gene and protein names is very often confusing in the whole manuscript and also figures. For example in Fig 4 only MPNxxx names are given but somewhere in the text its mentioned that these are the FtsZ/A homologs? And this goes on and on throughout the text. This is very confusing and I have to say that I know a lot about FtsZ but nothing about mpnxxx. I would suggest to use the homolog names when possible and maybe also name the mpn name in parentheses. Fortunately Lon and FtsH are mostly mentioned by these names

C Regulatory proteolysis? p6/7 Fig 2A The proteins which were observed to be downregulated upon Lon or FtsH depletion are not mentioned anymore and later not really discussed! But these identified proteins are interesting, they could for example be under the indirect control of regulatory proteolysis of an activator whose stability might be controlled by Lon or FtsH. Maybe this possibility could be examined or detected in the RNAseq data by observing differential changes in regions?

C In vivo degradation p7 To me there is no difference between measuring the relative abundance of proteins under the different depletion conditions by MS (Fig 2A, S4) or comparing the abundance of the same Flagtagged candidate substrate proteins by westernblot under the same conditions (Fig 3).

This is not a real "in vivo degradation assay", just because the abundance of a protein is compared at different conditions by westernblot or quantitative MS.

In contrast, the SILAC experiment is a much better for this purpose and a generally accepted way to measure real "in vivo" degradation, "protein half-lives" or stability, since it is a pulse-chase experiment, which can also specifically determine the subsequent change of protein levels by quantitative MS. The results of this experiment depicted Fig 2D suggests that there is a general correlation of the identified DD and DE candidates and their in vivo half-lives / degradation/ stability measured by the SILAC experiment.

However, I cannot see this protein by protein and compared in relation to the DD and DE candidate substrate proteins. To resolve this one could for example also state the respective "protein half-lives" measured by the SILAC experiment for the DD and DE candidate proteins listed in table S3. (As mentioned below, the supplementary information of the supplied excel files to me are not clear at all.) In addition, the title of the paragraph should be changed accordingly and the term "in vivo degradation" removed.

-Further questions (Q) and comments (C)

C Title the term "minimal genome organism" can be a little misleading or even confusing, since this organism has a relatively small genome that was not artificially reduced. Maybe just state the name of the organism in addition or instead of this term

Q p5 the last paragraph "Out of all the....cold stress adaptation" : For me this paragraph is a little confusing. It is not that clear what I learn from this text besides that some of these genes appear to behave like housekeeping genes and might also be involved in cold shock response? Is HrcA not a heat shock response regulator which indirectly responds to protein unfolding and misfolding? What is meant by exhibiting less variance and where do I see that in Fig S1?

Q Fig 1 Is the tet induced expression of Lon or FtsH similar to wt levels (Westernblots?) and is the growth of the respective strains comparable to wt cells? It would be informative to see the Westernblot experiments controlling the depletion of FtsH and Lon vs the respective complemented strains.

Q Fig 2A what time points during depletion vs complementation were measured and compared

Q p6 Is this protein-half-life measurement mentioned on p 6 the same SILAC time course experiment depicted in 2D?

C p6 In the supplementary files the excel sheets are called 103947_0_table_1679050_q5y38x.xlsx and 103947_0_table_1679052_q5y38x.xlsx and it is not clear, neither from the title nor from the text in the excel sheets, whether one of those excel sheets is table S1 or maybe S2. In the one of the excel sheets protein half-lives are listed, but also labeled true or false, which is quite confusing

Q p8 what is a restrictase?

Fig 3, 4 S6, S9 p7,8,9 Some of the presented westernblot experiments are not very conclusive since the addition of a flag tag appears to change the recognition mode of the possible C- or N-terminal degra. It is also possible that other putative cofactors, which could be necessary to identify substrate proteins for degradation in vivo, are inhibited by additional tags. Also, the mutational analysis of possible recognition loops is not too convincing.

Q p11 ...and FtsH would have a similar role in the control of the assembly of the Sec protein translocation complex. Where is that connection shown in the data?

Reviewer #3:

Summary:

This study by Burgos et al. investigates the cellular functions of the ATP-dependent proteases Lon and FtsH in *Mycoplasma pneumoniae*, in which both of these proteases are essential. The authors generated depletion strains and found that loss of Lon and FtsH results in severe growth defects. They went on and mapped proteome and transcriptome changes following FtsH and Lon depletion. Proteins whose levels were significantly increased following depletion were considered to be candidate substrates. Western Blot analyses for some of these candidate substrates confirmed increased protein levels in the absence of either of these proteases. For three of the validated Lon substrates the authors showed that degradation requires patches of hydrophobic amino acids at the termini. The authors also show that depletion of either protease results in global gene expression changes, and based on the analysis of pseudogene encoded proteins the authors suggest that Lon plays a role in protein quality control.

General comments:

Studying protein homeostasis and protein degradation in non-model organisms is an important research topic. In particular, the genome reduced bacteria *M. pneumoniae* offers an interesting system for studying the cellular roles of highly conserved bacterial proteases. While this study is generally well executed and provides new insights into the substrate pools of Lon and FtsH in *M. pneumoniae*, I find that it does not go particularly into depth. Experiments establishing the precise roles of FtsH and Lon in regulating the identified substrates are missing and the study provides only limited conceptual advance regarding the general functions of Lon and FtsH. My specific comments and suggestions to improve the manuscript follow below.

Major comments:

1. The authors measured protein abundance of putative substrates following protease depletion and called these assays "in vivo degradation assays" (Fig. 2). However, in order to monitor degradation in vivo, the authors need to conduct either pulse-chase assays or protein synthesis shut-down assays, in which protein decay is monitored over time following an antibiotic induced block of global protein synthesis (in wt and protease mutant). Ideally, to provide evidence that candidate substrates are direct protease substrates the authors would need to purify proteases and substrates and monitor degradation in vitro.
2. It would be interesting to see if the stabilization of some of the identified substrates contributes to the lethal phenotype of Lon and FtsH depletion. What are the phenotypic consequences of stabilization of the identified substrates in the presence of Lon and FtsH? What are the precise phenotypes of the FtsH and Lon depletion strains (with respect to cell division, DNA replication and cell envelope integrity), and does stabilization of the identified substrates explain these phenotypes?
3. The experiments shown in Fig. 4 make some attempts to address the mechanisms of substrate recognition. The authors made some interesting findings that specific motifs in the termini of three validated substrates are required for degradation. Here, the authors could have included additional mutations to pinpoint more precisely the minimal sequence motifs required for recognition. Has a similar analysis been done for FtsH? Are there key similarities or differences in how Lon and FtsH recognize their substrates?
4. The rationale behind the analysis of pseudogene-encoded protein stability was difficult for me to follow. What is the evidence that the gene products of these transcribed truncated genes are non-functional and misfolded? To support that Lon is involved in quality control, more direct experiments are required that assess Lon function under proteotoxic stress conditions. Does Lon depletion result in increased protein aggregation? Are elevated Lon levels required for heat tolerance? Does Lon associate with protein aggregates?
5. The discussion is in its current form lengthy and should be shortened to focus on the main

important points.

Minor comments:

1. The time points used in the depletion experiments should be specified in the figure or in the figure legend (all figures).
2. Proteins that have been experimentally validated to be Lon or FtsH substrates could be highlighted in Fig. 2F.

Reviewer #2:

The very interesting manuscript by Burgos et al investigate protein degradation in a genome reduced model organism, Mycoplasma pneumoniae, which encodes only two ATP dependent AAA+ proteases Lon and FtsH. Such AAA+ protease complexes are very important both for regulatory and general proteolysis, which is reflected by the apparent essentiality of lon and ftsH genes in this organism.

To study the effect of these genes on the transcriptome and proteome they constructed strains with lon or ftsH or both under the control of a Tetracycline controlled promoter, allowing the expression of these genes only in the presence of Tetracycline, and therefore also deplete the proteins encoded by these essential genes in the absence of tetracycline (Fig 1).

In the next paragraph on p6 the authors also mention that they measure proteome wide protein half-life times at 13 and 52h (Supposedly shown in the excel sheet which is probably Table S1?; and maybe related to Fig2D?).

Furthermore, to identify the possible protein substrate of the respective protease, they perform quantitative MS (Fig S4) and RNAseq experiments (Fig S11?) in Lon and FtsH depletion strains and correlate the observed transcriptional and protein abundance changes comparing the depletion condition with the complemented strains (Fig 2A). Here they observe upregulated proteins which they classify as differentially detected (DD, proteins not detected in the induced condition and detected in the depleted condition, and whose mRNA level did not increase more than 2-fold) or differentially expressed (DE, significant increase in abundance upon protease depletion that could not be attributed to an increase in mRNA levels). In addition, they also observe proteins which were downregulated upon Lon or FtsH depletion.

Interestingly when analyzing their candidate DD and DE proteins (Table S3) they observe separate and distinct putative substrate proteins, properties and characteristics, also due to the inherent different localization of FtsH as a membrane protein and the more cytosolic Lon (Fig 2 p7) And the cellular response to Lon vs FtsH depletion appears to be distinct (p9/10 Fig S11 or 12)

Then they go on and Flag tag specific candidate protein substrates to confirm again their degradation but the addition of the Flagtag itself is apparently interfering with the in vivo stability (Fig 3), which is well known in the field and also the characterization of possible motifs for the Lon recognition with this approach is not very conclusive (Fig 4)

The following analysis of Lon as a quality control protease is interesting and more

convincing (p 10 Fig 5) and the discussion is very interesting and insightful.

We thank reviewer 2 for the reviewing process and the valuable comments, which contributed to improve the quality of the manuscript. Below, we provide a detailed point-by-point response to the specific comments and questions.

-Major comments (C) and questions (Q)

C The usage of gene and protein names is very often confusing in the whole manuscript and also figures. For example in Fig 4 only MPNxxx names are given but somewhere in the text its mentioned that these are the FtsZ/A homologs? And this goes on and on throughout the text. This is very confusing and I have to say that I know a lot about FtsZ but nothing about mpnxxx. I would suggest to use the homolog names when possible and maybe also name the mpn name in parentheses. Fortunately Lon and FtsH are mostly mentioned by these names

We apologize for the inconveniences this may have caused. We have now corrected this throughout the text by using only the homolog gene name when possible. To provide a reference for the corresponding MPN gene number, we have also included this information in parentheses when first mentioned in the text as suggested. For genes/proteins for which there is no assignment to a known homolog gene, we used the MPN gene nomenclature followed in parenthesis by the putative gene annotation or function for reference. In the particular case of the hsdS subunits, in which there are several copies, we always mention both, common gene name (hsdS) and the specific MPN gene encoding the specific HsdS subunit. Additionally, we have amended the figures by adding the gene names when possible.

C Regulatory proteolysis? p6/7 Fig 2A The proteins which were observed to be downregulated upon Lon or FtsH depletion are not mentioned anymore and later not really discussed! But these identified proteins are interesting, they could for example be under the indirect control of regulatory proteolysis of an activator whose stability might be controlled by Lon or FtsH. Maybe this possibility could be examined or detected in the RNAseq data by observing differential changes in regions?

We thank reviewer 2 for this valuable suggestion and we agree that the analysis of down-regulated proteins could be also of interest. Following the same criteria we used to define candidate substrates, we found that out of the 40 and 20 proteins down-regulated after Lon and FtsH depletion, respectively, down-regulation of 19 and 11 proteins could not be explained by decreased mRNA expression, suggesting a translational or post-translational mechanism. These observations are now commented in the main text in P8, and the list of proteins can be found in Table EV2 (column of down-regulated candidates).

In addition, we have included a more detailed analysis of changes in expression after Lon and FtsH depletion focusing on known transcriptional regulators of *M. pneumoniae*, taking advantage of a recently published study in which gene regulation was comprehensively examined in this bacterium (Yus et al., 2019). This analysis is now included in the section of “Cellular response to Lon and FtsH depletion” in P12. In summary, only two indirect regulators (RecA and the lipoprotein MPN506) were found to be substrates of Lon and FtsH, respectively. Since perturbations in these indirect regulators were shown to induce only minor transcriptional changes (Yus et al., 2019), the majority of transcriptional changes observed upon Lon or FtsH depletion seem to be mediated by non-canonical factors or regulation of the activity of these transcriptional regulators as we propose for the HcrA transcriptional repressor. However, we did observe that half of the proteins down-regulated after Lon depletion are ribosomal proteins, and that these changes correlated with a decrease of the transcriptional activity of the operons encoding them (Fig. EV2, panel G). This transcriptional response could be the result of the moderate up-regulation of the WhiA-like repressor under Lon depleting conditions (yet WhiA did not pass our established criteria for being considered a substrate). These findings are now highlighted in the text.

C In vivo degradation p7 To me there is no difference between measuring the relative abundance of proteins under the different depletion conditions by MS (Fig 2A, S4) or comparing the abundance of the same Flagtagged candidate substrate proteins by westernblot under the same conditions (Fig 3).

This is not a real "in vivo degradation assay", just because the abundance of a protein is compared at different conditions by westernblot or quantitative MS.

In contrast, the SILAC experiment is a much better for this purpose and a generally accepted way to measure real "in vivo" degradation, "protein half-lives" or stability, since it is a pulse-chase experiment, which can also specifically determine the subsequent change of protein levels by quantitative MS. The results of this experiment depicted Fig 2D suggests that there is a general correlation of the identified DD and DE candidates and their in vivo half-lives / degradation/ stability measured by the SILAC experiment.

However, I cannot see this protein by protein and compared in relation to the DD and DE candidate substrate proteins. To resolve this one could for example also state the respective "protein half-lives" measured by the SILAC experiment for the DD and DE candidate proteins listed in table S3. (As mentioned below, the suppl Information of the supplied excel files to me are not clear at all.) In addition, the title of the paragraph should be changed accordingly and the term "in vivo degradation" removed.

We completely agree that protein abundances measured by MS and immunoblot

assays are similar complementary analyses, and therefore we were not presenting real “*in vivo* degradation assays”. We apologise for the misuse of the term. As mentioned, we do measure protein half-lives of individual *M. pneumoniae* proteins by SILAC-based proteomics. These data are presented now in Table EV1 (previous excel Table S1). As suggested, we have now included the specific protein half-lives for the candidate substrates in a separate column in excel Table EV3 (previous Table S3) and removed the term “*in vivo* degradation assay” from the title of the substrate validation section in P9. Furthermore, we have performed additional experiments regarding the substrate validation to clarify this section. By one hand, the previous immunoblot experiments showing expression of N-and C-terminal FLAG tagged substrates under inducing and depleting conditions (shown in previous Fig. 3) have been moved to Appendix Fig. S6 to show the effect of the position of the FLAG tag (see also comment below regarding FLAG interference). On the other hand, the previous Figure 3 (now Figure 4) has been improved by adding depletion time course experiments showing the stability of some substrates across different time points of depletion instead of a single time point. Additionally, we have monitored degradation of individual substrates by protein synthesis shut-down assays (see also response to comment 1 of referee 3).

-Further questions (Q) and comments (C)

C Title the term "minimal genome organism" can be a little misleading or even confusing, since this organism has a relatively small genome that was not artificially reduced. Maybe just state the name of the organism in addition or instead of this term

As suggested, we have modified the title as follows: “Protein quality control and regulated proteolysis in the genome-reduced organism *Mycoplasma pneumoniae*”

Q p5 the last paragraph "Out of all the....cold stress adaptation" : For me this paragraph is a little confusing. It is not that clear what I learn from this text besides that some of these genes appear to behave like housekeeping genes and might also be involved in cold shock response? Is HrcA not a heat shock response regulator which indirectly responds to protein unfolding and misfolding? What is meant by exhibiting less variance and where do I see that in Fig S1?

We agree that the paragraph was misleading and incomplete. We have now re-written this paragraph (now in P6) and modified the previous supplementary figures S1 and S2 (now a single figure EV1) to clarify the main results.

Our analysis of transcriptional changes under different perturbations suggest that Lon is the protease that differentially respond more upon different perturbations, whereas PepP, Lsp and PepF are the ones that exhibit less variability in expression, thus suggesting a

housekeeping like-behaviour. These findings are now shown in Fig. EV1 panel B. In particular, in the lower plot of panel B we show the standard deviation of the mRNA fold changes across the perturbations tested for each protease/chaperone gene, and we ranked them from the least to the most variable to facilitate visualization.

Additionally, we have now included (panel A) a clustered heat map showing the mRNA fold changes of each protease/peptidase gene for each perturbation. In the previous version of the manuscript we used data from 190 perturbations to perform the analysis. For clarity purposes, we have now used data from Yus et al., 2019 in which all the different perturbations were grouped in 35 unique conditions. As shown in Fig. EV1 panel A, glucose starvation and cold shock are the perturbations that induce the major changes across all proteases, and these observations are now highlighted in the text. Finally, we also comment about the specific regulation of proteases under the control of HcrA, showing that transcriptional changes of proteases and chaperones under the control of HcrA highly correlate (shown in Fig EV1, panel C).

In addition to the transcriptional analysis mentioned above, we now provide evidence that Lon depletion induces protein aggregation and significant transcriptional increases of genes regulated by the transcription factor HcrA. We propose that these changes may arise from the accumulation of misfolded proteins in the absence of Lon, resulting in the saturation of the GroEL/ES chaperonin system and subsequent inactivation of the HcrA repressor activity. These observations are discussed in P12 and P15.

Q Fig 1 Is the tet induced expression of Lon or FtsH similar to wt levels (Westernblots?) and is the growth of the respective strains comparable to wt cells? It would be informative to see the Westernblot experiments controlling the depletion of FtsH and Lon vs the respective complemented strains.

We have now included new Western blot experiments in Fig. 1 where protein levels of Lon and FtsH in wild-type cells are compared to conditional mutants grown under inducing or depleting conditions. As expected, the promoter replacement drives some variation in expression. In particular, Lon expression under inducing conditions is reduced compared to wild-type conditions, whereas FtsH induction promotes higher expression levels. Despite these differences, no significant impact on growth was observed. We believe that the assessment of the effect of Lon or FtsH depletion is better controlled by comparing the same mutant strain under inducing vs depleting conditions, rather than comparing with the wild-type strain, as differences in the genetic background may introduce confounding factors. On the other hand, as mentioned above, we have also included in the new version of the manuscript time course experiments monitoring over time Lon or FtsH depletion and the

respective changes in protein levels of candidate substrates. The text has been modified accordingly to these additions.

Q Fig 2A what time points during depletion vs complementation were measured and compared

The majority of experiments performed to assess the effect of Lon or FtsH (unless otherwise indicated) were done after 48h and 72h of depletion, respectively. This information is described in material and methods in the section “Culture conditions for Lon and FtsH depletion”. To further clarify the culture conditions for each experiment, we have specified this information in all figure legends.

Q p6 Is this protein-half-life measurement mentioned on p 6 the same SILAC time course experiment depicted in 2D?

Yes. We performed proteome wide measurements of protein half-lives in *M. pneumoniae*, which are reported individually for all proteins in Table EV1. In addition, as previously suggested, we have included the specific protein half-life for each candidate substrate in a separate column in Table EV3. In Figure 2D (now Fig. 3D) we show the statistical analysis of the distribution of protein half-lives among Lon and FtsH candidate substrates compared to other proteins. To clarify this, we have modified the text in P9 as follows:

“We also determined *M. pneumoniae* protein turnover rates by SILAC-based proteomics (Table EV1). Lon candidate substrates exhibited significant lower protein half-lives as compared to the average (Mann-Whitney-Wilcoxon [MWW] two-sided test, $p=6.53 \times 10^{-4}$; Fig. 3D and Table EV3).”

C p6 In the supplementary files the excel sheets are called 103947_0_table_1679050_q5y38x.xlsx and 103947_0_table_1679052_q5y38x.xlsx and it is not clear, neither from the title nor from the text in the excel sheets, whether one of those excel sheets is table S1 or maybe S2. In the one of the excel sheets protein half-lives are listed, but also labeled true or false, which is quite confusing

We apologize for this error. We assume that the file names have been unintentionally modified during the uploading process. According to the editorial format, we have now included in each excel table a separate tab called “README” in which we describe the different data presented in each column. In the particular case of Table EV1 (previous Table S1) in which we list the protein half-lives, the column stating true or false basically indicates

if the protein half-life for the specific protein has been clipped (True) or not (False) to a maximum of 300 hours. Some proteins exhibited a fitted apparent degradation rate very close to the dilution rate, implying that those proteins have very long half-lives. In order to simplify downstream analyses, we fixed the half-lives of those proteins to a maximum value of 300h, which reflected the smallest degradation rate measurable given the experimental variability. These details have been also specified in material and methods.

Q p8 what is a restrictase?

We used restrictase as a synonym of restriction enzyme. To clarify this, we have changed throughout the text the term “restrictase” by “restriction enzyme”.

Fig 3, 4 S6, S9 p7,8,9 Some of the presented westernblot experiments are not very conclusive since the addition of a flag tag appears to change the recognition mode of the possible C- or N-terminal degnon. It is also possible that other putative cofactors, which could be necessary to identify substrate proteins for degradation in vivo, are inhibited by additional tags. Also, the mutational analysis of possible recognition loops is not too convincing.

We agree that tags can influence protein stability. Unfortunately, specific antibodies against the validated candidate substrates are not available. For this reason, we tested constructs carrying both N or C-terminal tags. As shown now in Appendix Fig. S6 (data shown before in previous Fig. 3), at least one of the positions of the tag does not interfere with protein stability, allowing us to use these constructs for further analysis, including the mutational experiments of possible degnons. Even in those cases in which we did detect interference of the tag, we believe the result is relevant, as they suggested the position of possible degnons in some substrates (FtsA, FtsZ and DnaB). However, as the referee points out, it is also possible that the tag could interfere not only with a degradation signal, but also with the binding of putative cofactors required for degradation *in vivo*. We have now mentioned this possibility in the discussion.

Finally, we have also included in the revised manuscript a new mutational analysis assessing the contribution of single residues within the identified degnons (Fig. 5).

Q p11 ...and FtsH would have a similar role in the control of the assembly of the Sec protein translocation complex. Where is that connection shown in the data?

Our data indicate that SecE, SecD and SecY are FtsH substrates (Fig. 3F and Table EV3), suggesting that the degradation rate of these proteins could be influenced by the formation of stable complexes (discussed in P15). As we mention in the discussion, this

statement is supported by the fact that the stability of some components of the Sec pathway in *E. coli* depends on their assembly. Thus, it is likely that the same happens in *M. pneumoniae*. However, we agree that this fact is not directly demonstrated as we do for the assembly of the terminal organelle proteins. Of note, the components of the Sec translocation complex are essential, which make such experiments difficult to perform. In the revised manuscript we have tone down this statement as follows in P14:

“Also, we demonstrate that Lon can degrade unassembled components of the attachment organelle. We propose that FtsH may have a similar role in the control of the assembly of the Sec protein translocation complex.”

Reviewer #3:

Summary:

This study by Burgos et al. investigates the cellular functions of the ATP-dependent proteases Lon and FtsH in Mycoplasma pneumoniae, in which both of these proteases are essential. The authors generated depletion strains and found that loss of Lon and FtsH results in severe growth defects. They went on and mapped proteome and transcriptome changes following FtsH and Lon depletion. Proteins whose levels were significantly increased following depletion were considered to be candidate substrates. Western Blot analyses for some of these candidate substrates confirmed increased protein levels in the absence of either of these proteases. For three of the validated Lon substrates the authors showed that degradation requires patches of hydrophobic amino acids at the termini. The authors also show that depletion of either protease results in global gene expression changes, and based on the analysis of pseudogene encoded proteins the authors suggest that Lon plays a role in protein quality control.

General comments:

Studying protein homeostasis and protein degradation in non-model organisms is an important research topic. In particular, the genome reduced bacteria M. pneumoniae offers an interesting system for studying the cellular roles of highly conserved bacterial proteases. While this study is generally well executed and provides new insights into the substrate pools of Lon and FtsH in M. pneumoniae, I find that it does not go particularly into depth. Experiments establishing the precise roles of FtsH and Lon in regulating the identified substrates are missing and the study provides only limited conceptual advance regarding the general functions of Lon and FtsH. My specific comments and suggestions to improve the manuscript follow below.

We thank reviewer 3 for the reviewing process and the constructive feedback. We think that the additional experiments and analyses proposed have improved the quality of the manuscript. Please, see below a detailed point-by-point response to the specific comments.

Major comments:

1. The authors measured protein abundance of putative substrates following protease depletion and called these assays "in vivo degradation assays" (Fig. 2). However, in order to monitor degradation in vivo, the authors need to conduct either pulse-chase assays or protein synthesis shut-down assays, in which protein decay is monitored over time following an antibiotic induced block of global protein synthesis (in wt and protease mutant). Ideally, to provide evidence that candidate substrates are direct protease substrates the authors would

need to purify proteases and substrates and monitor degradation in vitro.

As we responded earlier to a similar comment raised by reviewer 2, we apologise for the misuse of the term “*in vivo* degradation assay”. We agree that MS and immunoblot analyses assessing protein abundances following protease depletion are similar complementary analyses, and therefore do not reflect real “*in vivo* degradation assays”. In the revised manuscript, we have included new experiments supporting the role of Lon and FtsH degrading some of the candidate substrates. These experiments are now shown in new Fig. 4 (previous Fig. 3). Accordingly, the results section: “Validation of Lon and FtsH substrates”; and the corresponding methods have been modified. By one hand, we performed depletion time course experiments instead of showing a single depletion time point, illustrating the association between protease expression and the stability of the candidate substrate. Since candidate substrates are expressed from an heterologous promoter, we believe it is very unlikely the existence of translational regulatory mechanisms promoting protein synthesis of these substrates in the absence of the protease. Therefore, protein degradation seems to be the most likely mechanism. To confirm this, we have also monitored degradation of the candidate substrates by protein synthesis shut-down assays as suggested. We would like to remark, however, the difficulties found in performing such experiments in *M. pneumoniae* as compared to other model organisms. By one hand, the majority of substrates tested are not detected in the presence of the protease, which jeopardizes the monitorization of protein degradation over time. In other organisms, this problem can be solved by transient overexpression of the substrate, typically using strong inducible systems. Unfortunately, to date, only the Tet inducible system is available in *M. pneumoniae*, and this system is already used during the construction of the conditional mutants. Additionally, in contrast to *E. coli*, *M. pneumoniae* cannot overexpress a protein by more than 2-3 fold. To overcome these problems, we first depleted Lon and FtsH, thus allowing sufficient detection of the substrates, and then we transiently induced the protease before blocking protein synthesis. Expression of the substrates at different time points after antibiotic treatment were then compared to non-induced cells. In general, these experiments allowed us to confirm the role of Lon or FtsH in the degradation of the selected substrates. However, the unusual long half-lives of the *M. pneumoniae* proteins (on average 68h), and the fact that cells need some time to recover after depletion of the protease make these assays not suitable to accurately measure protein half-lives. Additionally, the accumulation of other substrates or misfolded proteins (as we show for the Lon mutant) could saturate the protease capacity and therefore interfere with the degradation kinetics of the tested substrate. As an alternative and complementary approach, protein degradation was also assessed by pulse-chase SILAC based analysis in wild-type cells under normal growth conditions. Although this approach does not directly answer which

protease is responsible for the protein turnover, it allows to determine quantitatively the protein half-lives of individual proteins (half-lives listed in Table EV1). Therefore, we provide two complementary approaches: one approach that measures accurately protein half-lives (highlighting a correlation with the targets of Lon), and a second approach that monitors degradation of specific substrates to confirm the involvement of the protease.

2. It would be interesting to see if the stabilization of some of the identified substrates contributes to the lethal phenotype of Lon and FtsH depletion. What are the phenotypic consequences of stabilization of the identified substrates in the presence of Lon and FtsH? What are the precise phenotypes of the FtsH and Lon depletion strains (with respect to cell division, DNA replication and cell envelope integrity), and does stabilization of the identified substrates explain these phenotypes?

We agree that the identification of the factor or factors behind the essentiality of Lon and FtsH is interesting. However, whether the stabilization of a candidate substrate contributes to a specific phenotype is difficult to assess, as it depends on the identification of mutations or mechanisms that could protect the substrate from degradation. Unfortunately, this is difficult to perform in a systematic manner, especially if there are multiple degrons. In this regard, we have been able to stabilize the cell division factors FtsA, FtsZ and DnaB in the presence of Lon by adding a FLAG tag to the C-terminal (Appendix Figure S6) or introducing point mutations (new Fig. 5). Cells expressing these stable variants are viable with no obvious phenotypes, suggesting that lethality is not the result of the stabilization of these proteins and could be related to a more general role of the proteases in maintaining proteome homeostasis. To examine these possibilities, we have performed a transposon mutagenesis analysis to identify knock outs that could rescue the lethal phenotype (see Appendix Figure S14). However, this screening failed to identify mutants capable of surviving in the absence of Lon or FtsH. In particular, all the mutants isolated expressed Lon or FtsH in the absence of inducer, suggesting the accumulation of mutations derepressing the inducible system. These results suggest that the stabilization of a single non-essential protein is unlikely to explain the lethal phenotype as it occurs in *E. coli*, in which the essentiality of FtsH seems to be associated with its role in regulating LpxC. One limitation of the screening, however, is that essential genes cannot be disrupted and therefore assessed for its role in the lethal phenotype. These observations are now discussed in P16.

In addition, in the revised manuscript we added new experiments examining the phenotypes of Lon and FtsH depletion strains, highlighting the role of these proteases in maintaining proteome homeostasis. As a result, a new Figure (Fig. 2), a new result section named “Phenotypic characterization of Lon and FtsH mutants”, and the corresponding

methods has been included in the revised manuscript. In particular, we performed new growth curve analysis measuring DNA and protein cell biomass, showing that depletion of Lon and FtsH results in DNA and protein synthesis inhibition (Fig. 2A). Defects on DNA replication were further confirmed by pulse chase experiments using the analog bromodeoxyuridine (Appendix Figure S3). In the case of depletion of Lon, we provide transcriptome and proteome data indicating a decrease in expression of ribosomal proteins, which is consistent with the observed protein synthesis inhibition (Fig. EV2 and Table EV2). As suggested by the reviewer, we also performed new experiments assessing cell envelope integrity in both mutants (Fig. 2C). We found that FtsH depleted strains exhibited cell membrane damage under normal and mildly membrane disruptive conditions, a phenotype not observed for Lon depleted strains. In fact, we were unable to regrow FtsH mutant cells following FtsH depletion, suggesting important cellular damage. These results are consistent with the membrane localization of FtsH, and that the majority of substrates identified are membrane associated. Thus, FtsH seems to play a critical role in maintaining general membrane protein homeostasis in this reduced genome organism. In contrast, Lon depletion was associated with increased protein aggregation (Fig 2B), suggesting a role in general protein quality control (see also comment 4 below). Although we cannot discard that the stabilization of a specific substrate could contribute to the lethal phenotype dysregulating essential processes, overall these results also agree with a general role of these proteases in proteome homeostasis, which can explain their essentiality. These observations are now discussed in the revised version of the manuscript.

3. The experiments shown in Fig. 4 make some attempts to address the mechanisms of substrate recognition. The authors made some interesting findings that specific motifs in the termini of three validated substrates are required for degradation. Here, the authors could have included additional mutations to pinpoint more precisely the minimal sequence motifs required for recognition. Has a similar analysis been done for FtsH? Are there key similarities or differences in how Lon and FtsH recognize their substrates?

Following the recommendation of the reviewer, we have now included additional analysis examining the effect of single point mutations within the degrons identified in FtsA, FtsZ and DnaB. Although mutations of individual residues in FtsA do not seem to affect stability, we identified single point mutations in FtsZ and DnaB degrons that are sufficient to prevent degradation (in particular, aromatic amino acids). These results suggest that hydrophobic sequences as short as a single residue can likewise promote Lon-dependent degradation, although in other cases such as in FtsA several residues may be required. We believe that this is an important contribution as it highlights the broad capacity of the Lon protease to degrade a protein, and that other underlying mechanisms such as the regulation of

the accessibility of these regions are key to regulate the stability of substrates. It is especially interesting to note how a genome-reduced organism, lacking protein adaptors and other ATP-dependent proteases, has evolved to take advantage of Lon to regulate its proteome. As a result of the addition of these new analyses, Fig 5 (previous Fig. 4) and the corresponding text have been modified accordingly.

As mentioned at the beginning of the results section: “Identification of protease recognition determinants”, we were unable to identify common features within Lon or FtsH substrates that could suggest degradation motifs. This is consistent with previous reports showing that recognition mechanisms of known Lon and FtsH substrates are highly diverse (Tsilibaris et al., 2006; Bittner et al, 2017), probably influenced by structural constraints affecting the accessibility of the degrons. Therefore, identification of specific degrons within substrates requires individual and extensive mutational analysis for each substrate. In our study, we could identify degrons for three Lon substrates due to the observation that tag fusions prevented degradation. These results allowed us to narrow down possible degrons in these proteins. Unfortunately, a similar analysis was not feasible for FtsH substrates due to lack of information. However, we agree that these analyses could be interesting in future studies by performing random mutagenesis.

4. The rationale behind the analysis of pseudogene-encoded protein stability was difficult for me to follow. What is the evidence that the gene products of these transcribed truncated genes are non-functional and misfolded? To support that Lon is involved in quality control, more direct experiments are required that assess Lon function under proteotoxic stress conditions. Does Lon depletion result in increased protein aggregation? Are elevated Lon levels required for heat tolerance? Does Lon associate with protein aggregates?

We hypothesized that truncated variants present in the genome are probably not correctly folded, thus increasing the likelihood to be Lon substrates. However, we agree with the reviewer that we do not have direct evidence that all the truncated genes detected in *M. pneumoniae* are non-functional or misfolded. It is possible that in some cases, these truncated variants are still functional or even may have acquired new functions. Although we are aware of this limitation, our analysis suggests there is a significant proportion of these variants whose expression is Lon-dependent. As an example, we validated the truncated variant MPN304 (ArcANt), now through depletion time course experiments and *in vivo* degradation assays (new Fig 6B, previous Fig 5A). In this case, there is reported evidence that the metabolic pathway in which MPN304 is involved is inactive (Rechinitzer et al, 2013). To further support the role of Lon in degrading unfolded proteins, we have included in the revised manuscript an additional experiment in which we assessed the stability of an unstable

luciferase reporter variant, and show that it depends on Lon expression. This new evidence is shown in panel A of Fig. 6 and described in the results section: “Role of Lon as a quality control protease in *M. pneumoniae*”. The corresponding methods have been updated accordingly. Additionally, during the revision of our manuscript we noticed that MPN109 and MP110 split genes were not included in our list of possible truncated variants. As a result, we have updated the table EV4 and the resulting analysis in P12-13. The addition of these two truncated variants does not modify the conclusions.

As suggested, we have also performed additional experiments to support the role of Lon in quality control. By one hand, we provide evidence of the role of Lon to cope with proteotoxic stress conditions. In particular, we show that Lon depleted cells exhibit increased sensitivity to heat stress as compared to Lon expressing cells (shown in new Fig. 2D). Unfortunately, we could not assess the role of FtsH, as we were unable to regrow the mutant cells after FtsH depletion. These observations are now discussed in the new result section “Phenotypic characterization of Lon and FtsH mutants”.

As mentioned before we also performed new experiments to assess whether Lon or FtsH depletion results in increased protein aggregation. Consistent with a role of Lon in degrading misfolded or damaged proteins, we detected an increase in protein aggregates in the absence of Lon under normal growth conditions. This phenotype was magnified when Lon mutants were exposed to heat stress. These results are shown in new Fig. 2B, and discussed in the new result section “Phenotypic characterization of Lon and FtsH mutants”. Furthermore, we performed MS analysis of the insoluble fractions, revealing an enrichment of Lon substrates in Lon depleted cells. These results suggest that Lon substrates tend to aggregate when accumulated. To show these results, a new panel (panel C) has been included in Fig. 6 (previous Fig. 5), and the results section: “Role of Lon as a quality control protease in *M. pneumoniae*“ modified accordingly. Also, the corresponding methods have been included in the revised manuscript and the new protein dataset uploaded to the PRIDE repository under the identifier PXD021506.

Following the reviewer suggestions, we further examined the expression of Lon under heat stress and its partitioning between soluble and insoluble fractions. After heat treatment, we observed a clear shift of Lon into the insoluble fraction, suggesting association of Lon with protein aggregates. These new results are shown in a new panel (panel D) in Fig. 6 (previous Fig. 5), and described in the results section: “Role of Lon as a quality control protease in *M. pneumoniae*“.

5. The discussion is in its current form lengthy and should be shortened to focus on the main important points.

The previous discussion section has been shortened in the revised version, and it has been modified focusing on the new findings.

Minor comments:

1. The time points used in the depletion experiments should be specified in the figure or in the figure legend (all figures).

We have now included the depletion time points used for each experiment in all figure legends.

2. Proteins that have been experimentally validated to be Lon or FtsH substrates could be highlighted in Fig. 2F.

Following this recommendation, we have highlighted in bold the validated substrates in previous Fig. 2F (Fig. 3F in the revised version).

Thank you for sending us your revised manuscript. We have now heard back from the two reviewers who were asked to re-assess your study. You will see from the comments below that both reviewers are overall positive and support publication of the article in *Molecular Systems Biology*. However, the reviewers also raised a couple of minor issues that still need to be addressed. In particular, Reviewer #3 is concerned about the observed long protein half-lives under Lon inducing conditions, which needs to be carefully discussed and explained.

On a more editorial level, we would ask you to address a few remaining editorial issues listed below

Reviewer #2:

The revised manuscript "Protein quality control and regulated proteolysis in the genome-reduced organism *Mycoplasma pneumoniae*" by Burgos et al with its new experiments, revised text, figures and tables has improved a lot and gives new insight into the essential and mostly complementary roles of Lon and FtsH, the sole AAA+ proteases in this organism, in protein and membrane homeostasis.

I have just some questions and comments

-I assume using the SILAC pulse-chase approach to study in vivo protein degradation was not applied, because of the difficult growth behavior of the Lon or FtsH depletion strains, as described in the response letter?

-Fig 6D It would be great to know, which of the identified Lon substrates were found in the aggregated proteins. Maybe it could be possible to indicate them for example in Table EV3.

-HrcA/MPN124 In about half of the manuscript and some of the figures, hrcA is misspelled as hcrA. The same is actually true for the often referred to Yus et al publication.

- p12 "Thus, upregulation of WhiA could explain the decrease of ribosomal proteins under Lon

depleting conditions." Is WhiA a repressor and are higher levels of WhiA detected in the proteome data?

-p14 "Pip and the ClpB chaperone, are essential, with Tig and Pip being fitness (Lluch-Senar et al, 2015)." What does this sentence mean?

-p16 "...as we have shown with an engineered unstable reporter protein." is this the mentioned Firefly Luciferase, and if so why is it engineered to be unstable??? Firefly Luciferase is a multidomain folded protein (similar to non-ribosomal peptide synthetases), which is known to easily misfold...

-p16 "For example, proteins related to recombination and DNA repair were upregulated in the absence of Lon, including RecA (MPN490), RrlB sigma accessory protein (MPN534, (Torres-Puig et al, 2018)), and DNA polymerase IV (MPN537)" Is this "upregulated" because of an indirect effect of Lon on the regulation or were "higher levels detected" because of a direct effect of Lon?

Reviewer #3:

The authors have carefully revised their manuscript and added a number of experiments to address the reviewers' comments.

The validation of candidate substrates is more convincing in the new version of the manuscript. I like in particular the time course experiments of Lon depletion in combination with the RNA-seq data (Figs. 4A-B). The degradation assays shown in Fig. 4C, 4F, 4I are less convincing in my eyes. It is unexpected that the protein half-lives are so extremely long under Lon inducing conditions, much longer than the half-lives measured for the same substrates by SILAC, which makes me wonder if the synthesis shut-down assays have worked as they should.

As the authors have decided to show these assays, they should describe and discuss the results in the main text, which is currently only partially the case. In particular, the data shown in Fig. 4C are not sufficiently described. The sentence "In all three cases, the expression of the N-terminal FLAG fusion was dependent on Lon-mediated degradation, suggesting these cell division factors are Lon regulated." is ambiguous and should be changed to more correctly describe the results in Figures 4B and 4C. A short discussion of the results should also be included to provide explanations for the unexpected long protein half-lives under Lon inducing conditions.

Finally, information about the reproducibility of these data is missing. The authors should indicate the number of experimental replicates and either include quantifications of band intensities with standard error, or alternatively, state whether the presented Western blots are representative for all replicates. Technically this applies to all Western blots shown in this manuscript, but is particularly important in cases where the Western blots show small changes in protein levels.

Responses to reviewers:

We thank the reviewers for revising our manuscript again and for the additional and constructive comments.

Reviewer #2:

The revised manuscript "Protein quality control and regulated proteolysis in the genome-reduced organism *Mycoplasma pneumoniae*" by Burgos et al with its new experiments, revised text, figures and tables has improved a lot and gives new insight into the essential and mostly complementary roles of Lon and FtsH, the sole AAA+ proteases in this organism, in protein and membrane homeostasis.

I have just some questions and comments

-I assume using the SILAC pulse-chase approach to study in vivo protein degradation was not applied, because of the difficult growth behavior of the Lon or FtsH depletion strains, as described in the response letter?

As the reviewer indicates, the special growth conditions and the growth impairment of the mutants after protease depletion unfortunately make these experiments difficult to perform. In addition, constant cell growth rate is a critical requirement for the precise estimation of the protein half-lives in the SILAC pulse-chase method, in order to take into account the dilution of proteins due to cell growth. Such a condition is not satisfied after protease depletion, where we observed a slowdown of cell growth.

-Fig 6D It would be great to know, which of the identified Lon substrates were found in the aggregated proteins. Maybe it could be possible to indicate them for example in Table EV3.

We have added this information in a new column in Table EV3 as suggested.

-HrcA/MPN124 In about half of the manuscript and some of the figures, hrcA is misspelled as hcrA. The same is actually true for the often referred to Yus et al publication.

We appreciate very much to bring to our attention this error. It is now corrected.

- p12 "Thus, upregulation of WhiA could explain the decrease of ribosomal proteins under Lon depleting conditions." Is WhiA a repressor and are higher levels of WhiA detected in the proteome data?

Yes. The fact that WhiA is a repressor and higher protein levels are detected under Lon depleted conditions is stated in p12 as follows:

“However, the WhiA-like repressor (MPN241), which has been associated with repression of the main ribosomal protein operon in *M. pneumoniae* (Yus *et al*, 2019), and the membrane anchored PrkC kinase (MPN248) were moderately upregulated (0.78 and 0.98 log₂, respectively) at the protein level after Lon or FtsH depletion, respectively.”

-p14 "Pip and the ClpB chaperone, are essential, with Tig and Pip being fitness (Lluch-Senar *et al*, 2015)." What does this sentence mean?

We intended to point out that transposon essentiality studies indicate that all the components of the protease and chaperone systems are essential for cell survival, except Tig, Pip and ClpB. In addition, despite transposon insertions can be detected in Tig and Pip encoding genes, these genes are classified in an intermediate essentiality category defined as fitness. The fitness category includes those genes that under certain conditions have a detrimental impact on growth but are not essential (see Lluch-Senar *et al.*, 2015 for reference). In order to clarify this sentence, we have modified the text in p14 (now p15) as follows:

“Transposon essentiality studies indicate that all proteins involved are essential for cell survival under normal growth conditions, except for the trigger factor (Tig, MPN331), Pip and the ClpB chaperone. Despite *tig* and *pip* encoding genes are non-essential, they are classified as fitness genes, meaning that transposon insertions in these genes negatively impact but not prevent growth (Lluch-Senar *et al*, 2015)”.

-p16 "...as we have shown with an engineered unstable reporter protein." is this the mentioned Firefly Luciferase, and if so why is it engineered to be unstable??? Firefly Luciferase is a multidomain folded protein (similar to non-ribosomal peptide synthetases), which is known to easily misfold...

Although we are aware of the low thermal stability of the luciferase protein, this reporter system has been broadly used in many organisms including *M. pneumoniae* (Weber *et al.*, 2020), showing certain stability and activity in the culture conditions tested. Here, we attempted to engineer a fully unstable variant to better see the effect of Lon depletion. For this, as it is described in Material and Methods we used the protein design software FoldX, which has been broadly used to predict stabilizing and destabilizing mutations. To clarify to which unstable protein we were referring in the sentence mentioned in p16, we have modified the sentence as follows: “...as we have shown with an engineered unstable variant of the luciferase reporter.”

-p16 "For example, proteins related to recombination and DNA repair were upregulated in the absence of Lon, including RecA (MPN490), RrlB sigma accessory protein (MPN534, (Torres-Puig *et al*, 2018)), and DNA polymerase IV (MPN537)" Is this "upregulated" because

of an indirect effect of Lon on the regulation or were "higher levels detected" because of a direct effect of Lon?

RecA, RrlB and DNA polymerase IV were indeed classified as candidate Lon substrates. RNA-seq data indicate that the observed protein upregulation of these proteins is not due to indirect transcriptional changes induced by other factors affected by Lon. Therefore, upregulation of these proteins is probably a direct effect of Lon depletion. To clarify this, we have reworded the mentioned sentence (now in p17) as follows:

“For example, proteins related to recombination and DNA repair were identified as Lon candidate substrates, including RecA (MPN490), RrlB sigma accessory protein (MPN534, (Torres-Puig *et al*, 2018)), and DNA polymerase IV (MPN537) were identified as Lon candidate substrates.”

Reviewer #3:

The authors have carefully revised their manuscript and added a number of experiments to address the reviewers' comments.

The validation of candidate substrates is more convincing in the new version of the manuscript. I like in particular the time course experiments of Lon depletion in combination with the RNA-seq data (Figs. 4A-B). The degradation assays shown in Fig. 4C, 4F, 4I are less convincing in my eyes. It is unexpected that the protein half-lives are so extremely long under Lon inducing conditions, much longer than the half-lives measured for the same substrates by SILAC, which makes me wonder if the synthesis shut-down assays have worked as they should.

As the authors have decided to show these assays, they should describe and discuss the results in the main text, which is currently only partially the case. In particular, the data shown in Fig. 4C are not sufficiently described. The sentence "In all three cases, the expression of the N-terminal FLAG fusion was dependent on Lon-mediated degradation, suggesting these cell division factors are Lon regulated." is ambiguous and should be changed to more correctly describe the results in Figures 4B and 4C. A short discussion of the results should also be included to provide explanations for the unexpected long protein half-lives under Lon inducing conditions.

Finally, information about the reproducibility of these data is missing. The authors should indicate the number of experimental replicates and either include quantifications of band intensities with standard error, or alternatively, state whether the presented Western blots are

representative for all replicates. Technically this applies to all Western blots shown in this manuscript, but is particularly important in cases where the Western blots show small changes in protein levels.

As we described in our previous response letter we could not perform standard *in vivo* degradation assays, in part because the conditional mutant system used in the study made the substrate monitoring difficult. In addition, *M. pneumoniae* exhibits long protein half-lives as compared to other bacterial systems (68h on average). In fact, SILAC measurements indicate that even unstable substrates such as FtsA and FtsZ exhibit relatively long half-lives of 4h and 6h, respectively. Fig 4C shows for example that depletion of FtsA and FtsZ starts around 9h (FtsA) to 6h (FtsZ), which is not so different from the SILAC estimates. However, we agree that our adapted “*in vivo* degradation assay” has several limitations and therefore it can only be considered as a qualitative method to assess the role of the protease. We believe that the SILAC pulse-chase method is much more appropriate to determine precisely the protein degradation kinetics. To clarify this, we have described in more detail the results of these experiments and discussed the limitations of the degradation assay in the main text in p9-10. Also, we have stated in the corresponding figure legends that the data presented are representative of two independent experiments.

Thank you again for sending us your revised manuscript. We are now satisfied with the modifications made and I am pleased to inform you that your paper has been accepted for publication.

YOU MUST COMPLETE ALL CELLS WITH A PINK BACKGROUND

Corresponding Author Name: Luis Serrano
 Journal Submitted to: Molecular Systems Biology
 Manuscript Number: MSB-20-9530